# Using Absorption Models for Insulin and Carbohydrates and Deep Leaning to Improve Glucose Level Predictions

**DOI:** 10.3390/s21165273

**Published:** 2021-08-04

**Authors:** Laura Martínez-Delgado, Mario Munoz-Organero, Paula Queipo-Alvarez

**Affiliations:** 1School of Engineering, Universidad Carlos III de Madrid, 28911 Leganés, Madrid, Spain; 100316077@alumnos.uc3m.es; 2Department of Telematic Engineering and UC3M-BS Institute of Financial Big Data, Universidad Carlos III de Madrid, 28911 Leganés, Madrid, Spain; 3Department of Telematic Engineering, Universidad Carlos III de Madrid, 28911 Leganés, Madrid, Spain; pqueipo@it.uc3m.es

**Keywords:** deep learning, LSTM, data processing, glucose level prediction, insulin absorption, carbohydrate absorption, machine learning models

## Abstract

Diabetes is a chronic disease caused by the inability of the pancreas to produce insulin or problems in the body to use it efficiently. It is one of the fastest growing health challenges affecting more than 400 million people worldwide, according to the World Health Organization. Intensive research is being carried out on artificial intelligence methods to help people with diabetes to optimize the way in which they use insulin, carbohydrate intakes, or physical activity. By predicting upcoming levels of blood glucose concentrations, preventive actions can be taken. Previous research studies using machine learning methods for blood glucose level predictions have mainly focused on the machine learning model used. Little attention has been given to the pre-processing of insulin and carbohydrate signals in order to mimic the human absorption processes. In this manuscript, a recurrent neural network (RNN) based model for predicting upcoming blood glucose levels in people with type 1 diabetes is combined with several carbohydrate and insulin absorption curves in order to optimize the prediction results. The proposed method is applied to data from real patients suffering type 1 diabetes mellitus (T1DM). The achieved results are encouraging, obtaining accuracy levels around 0.510 mmol/L (9.2 mg/dl) in the best scenario.

## 1. Introduction

Type 1 diabetes is among the most frequent chronic diseases in children and young people, having an important social and health impact and directly affecting the quality of life of the people who suffer from it. The disease begins when the pancreas does not produce enough insulin which results in an increase in the level of glucose in the blood stream and can cause heart and kidney problems or damage some of the nerves in the body. Despite the latest advances, an exact mechanism for treating the disease is not known. This increases the importance of proposing and validating techniques that facilitate the monitoring and treatment of type 1 diabetics [1] and even the development of systems that allow the identification of patterns that can help finding the best treatment for each patient. Machine learning techniques and algorithms are proving to be able to build a solid foundation in many areas in general and for blood glucose prediction in particular [2,3]. The use of data from real patients combined with machine learning techniques and prediction algorithms allows us to monitor patients, predict possible drops or rises in blood glucose levels, and establish personal recommendations based on their particular and personal data. In order for the recommendations to be accurate, an appropriate combination of machine learning models and good quality data are required. Currently, both theoretical models [4] and machine learning models (in particular deep learning based models [3]) are being successfully applied to data obtained from blood glucose monitors (continuous glucose monitors or CGM) together with manually provided information, such as insulin bolus and meals. The sparsity of the data in both the insulin injection and meal intake signals represent a challenge for the convergence of machine learning models in general, and more especially for deep learning models in particular. A proper combination of machine learning models and theoretical physiological absorption models for insulin and carbohydrates could improve convergence and results for deep learning models. This paper uses a recurrent neural network (RNN) based on LSTM cells in order to estimate future levels of blood glucose based on past readings coming from a continuous blood glucose monitor (CGM), insulin injections, and carbohydrate intake and works on different absorption models to process the data available from real patients to obtain the best prediction of their next glucose levels and, thus, prevent possible rises or falls in the blood sugar level.

## 2. State-of-the-Art

Both insulin injections and carbohydrate intake have an important influence in blood glucose levels for type 1 diabetes mellitus (T1DM) patients. An optimal regulation will require a proper combination of timely injections of insulin boli, taking into account the amount of insulin units and the type of insulin to be used, to mitigate the timely effect caused by the exact amount of carbohydrates taken depending on the current and past levels of blood glucose. Several absorption models for insulin and carbohydrates have been proposed inside blood glucose regulation models in order to assess the influence over time of human actions (meal intake or insulin injections) into plasma concentration levels and, therefore, provide an estimation of the influence in blood glucose levels. Chiara et al. [5] proposed a simulation model in normal humans, not suffering type 1 diabetes mellitus (T1DM), that describes the physiological events that occur after a meal. The parameters for the model were set to fit the mean data of a large normal subject database that underwent a triple tracer meal protocol which provided quasi-model-independent estimates of major glucose and insulin fluxes. A parametric model was developed by decomposing the system into subsystems. Wilinska et al. [6] proposed a set of theoretical models for insulin lispro kinetics with bolus and continuous subcutaneous insulin infusion (CSII) modes of insulin delivery. Eleven alternative models of insulin kinetics were proposed. The models took into account several absorption configurations, taking into account different rates of absorption, and considering the influence of the volumes of insulin. The models also studied two pathways for insulin absorption (fast and slow) and were based on compartment divisions. Absorption models for insulin and carbohydrates can be expressed in mathematical terms in the form of differential equations. A particular mathematical representation can be found in Ruan et al. [4]. The authors developed a new hierarchical model to relate subcutaneous insulin delivery and carbohydrate intake to continuous glucose monitoring over 12 weeks while describing day-to-day variability. A compartment model was proposed which comprised five linear differential equations. Hovorka et al. [7] also proposed a compartment model, which represented the gluco-regulatory system and included submodels representing absorption of subcutaneously administered short-acting insulin Lispro and gut absorption, showing promising results. Hajizadeh et al. [8] presented a plasma insulin concentration (PIC) estimator that incorporated Hovorka’s glucose-insulin model [7] with the unscented Kalman filtering algorithm. The authors took interest in improving convergence. Haiya et al. [9] modeled the insulin therapies using a delay differential equation model. The authors studied the dynamics of the model both qualitatively and quantitatively. An explanatory review about early mathematical models designed to understand the pathogenesis of diabetes is captured in [10].

Machine learning techniques have also been used as an alternative method in order to estimate upcoming values of blood glucose (BG) levels. Machine learning methods are able to learn from the experience (using labeled training data) how to combine the information in the input glucose, carbohydrate, and insulin signals in order to minimize the errors in the prediction of upcoming BG levels (and, therefore, training a black box model that will simulate the underlying physiological model). Pappada et al. [11] used a neural network-based (NN) model in order to estimate future values for blood glucose levels. The input signals for the model were food intake, insulin injections, and physical activity data. The author found that the proposed NN model worked well for anticipating hyperglycemic episodes (achieving accuracy values around 0.95 for a 60 min prediction horizon), but failed to predict hypoglycemic episodes. The authors in [12] proposed a machine learning based model that could be trained for blood glucose prediction based on the use of a Kalman filter in order to estimate hidden values in the model. A support vector regression (SVR) algorithm was used to estimate upcoming values for BG levels taking into account the current and past levels of carbohydrates, insulin, and BG levels. The authors found that results were similar to those manually predicted by a doctor. The machine learning model proposed by the authors in [13] was also based on an SVR model that, using the information provided by a single continuous glucose monitoring (CGM) device, tries to predict blood glucose levels independently of other factors. The results were able to improve those of other similar previous studies by adding differential evolution (DE) algorithms over data from 12 patients using CGM devices. The authors obtained average values for the root mean square error (RMSE) of 10.78 and 12.95 mg/dL for prediction horizons (PHs) of 30 and 60 min. Another machine learning model based on artificial neural networks (ANNs) for the blood glucose level prediction of type 1 diabetes (T1D) was proposed by Ali et al. [14]. The authors proposed an improved method based on using only previous glucose levels as inputs. The results were validated on real CGM data of 13 patients, achieving RMSE values of 7.45 mg/dL and 9.03 mg/dL for prediction horizons (PHs) of 30 and 60 min. A meta-learning approach was proposed in [15]. The authors proposed the idea of using regularized learning algorithms in predicting blood glucose. Meta-learning approaches are designed to be portable from patient to patient while outperforming other algorithms in terms of clinical accuracy.

Recently, some deep learning algorithms have also been applied for predicting upcoming BG levels trying to achieve a better performance compared to previously shallow methods. The authors in [16], for example, described some methods for deep multi-output blood glucose forecasting. The results were validated using deep learning methods. The authors showed that deep learning models outperformed previous shallow learning alternatives. The study in Mhaskar et al. [17] was also based on a deep learning approach in order to estimate upcoming BG levels. The study only used the previous BG levels in order to estimate future BG level values but using a pre-clustering mechanism to train specific models for hypo, eu, and hyperglycemic segments. Again, the authors showed that deep learning methods were able to outperform shallow network configurations.

In this manuscript, we propose and validate a new mechanism that incorporates physiological absorption models for insulin and carbohydrates into the pre-processing steps of a machine learning algorithm using a recurrent neural network based on LSTM cells. The learning algorithm is then able to learn the intricate dependencies between blood sugar levels and estimated values for blood insulin and carbohydrate concentrations trying to make it simpler for the model to learn.

## 3. Materials and Methods

To validate the methods proposed, the D1NAMO dataset [18] has been used. It is a multi-modal dataset for research on non-invasive type 1 diabetes management. It contains data from 29 real participants, 20 of them used as healthy controls, and the other 9 patients with type 1 diabetes. The participants were monitored in the same way using the Zephyr BioHarness 3 wearable chest-belt device, recording information on the heart and respiration rates, and body acceleration. A continuous glucose monitoring (CGM) device was used to record glucose levels. The participants also recorded meals and insulin doses. Part of the information in the dataset will be used to train the machine learning models proposed in this paper and the rest of the data will be used for validating the results.

The information in the D1NAMO dataset [18] is likely not to be complete. Meals and insulin injections were manually recorded by the participants and it is likely that some of the data values for some meals and insulin injections are missing. Several studies have highlighted the importance of having accurate data for meals. The study in [19] analyzed CGM postprandial data and using fuzzy logic the authors were able to detect the majority of meals and many of the snacks. The authors in [20] also used a set of rules to detect incorrect meal records. As a future work, we plan to use machine learning techniques to better detect missing or incorrect information in the annotated meals. Table 1 captures the number of meals and insulin injections recorded by patient. The table also provides information about some incidences that have been observed in the meal information available. In order to get the best information to validate the methods proposed in this manuscript, only patients with IDs 1, 2, 4, 6, 7, and 8 have been used. Participant number 3 only recorded 3 meals and 3 insulin injections in 4 days and it is likely that most of the values are missing. Participant 5 did not provide information about the dates and times in which the meals were eaten. Participant 9 was also discarded since some of the information was manually assessed as missing.

There were 9 patients with normal outpatient lifestyles. Patients 003, 005, and 009 were not included in the study due to incompleteness in the meals and CGM data. Because of that, only 6 patients were included. Extended information about the relevant patients is shown in Table 2.

The individual glucose statistics for the 9 patients are captured in Figure 6 in [18].

### 3.1. Data Pre-Processing

The goal of the method proposed in this manuscript is to estimate the blood glucose (BG) level of patients using a prediction horizon (PH) of 60 min. For clarity, throughout this paper the unit mmol/L is used for blood glucose prediction. PHs of 30 and 60 min are common in previous research studies [2,14]. The latest 24 samples from the CGM device (2 h of data) are combined with the information from the insulin injections and carbohydrate intakes, together with the estimation of the physical activity, in order to estimate the value expected for the output of the CGM device with a prediction horizon of 60 min (12 samples) in advance. In order to combine the information from the different input signals, a data pre-processing module has been added to the proposed method. It is important to this subsection to describe the details for the pre-processing steps.

#### 3.1.1. Glucose Data Pre-Processing

The BG signal is recorded from a continuous glucose monitoring (CGM) device providing an update every 5 min. The dataset provides the information for each participant in different files. The samples taken from the CGM device are organized in a different file per day. The BG signal pre-processing module will join the information in the different files and perform clock synchronization to a canonical time frame providing samples at a constant rate every 5 min.

Data are processed for each patient separately. By processing the data of patient 001, the signal from Figure 1 is obtained.

After processing, glucose level values are obtained every 5 min for each patient. In order to fill in the training and validation subsets, two hour windows of data (24 values every 5 min from t-23 to t for every instant of time t) will be used to feed the input of a machine learning algorithm and the value after one hour (t + 12) will be used as the desired output. A sliding window mechanism will go through the entire BG signal for each participant and randomly assign each data window to the training or validation subsets.

#### 3.1.2. Acceleration Data Pre-Processing

Acceleration data measures acceleration recorded from the accelerometer sensor worn by each patient. It is used to estimate if the patient performed physical activity in a time interval. Physical activity increases the consumption of blood glucose and has an impact on future values expected for the BG signal. The higher the intensity and duration of physical exercises, the lower the future glucose levels.

The acceleration data measured by the Zephyr Bioharness 3 device is used to estimate if the patient performed physical activity in a time interval. The acceleration is measured along 3 movement axis: vertical, lateral, and sagittal. This device has only been used during the day (the participants removed the band during the night for their convenience and comfort). The data pre-processing module only used fragments of data in which all the input signals were present in the dataset. Therefore, the segments during the night in which there was no information from the accelerometer were removed from the training and validation subsets.

The acceleration data sensor collects a sample every 10 ms. It is measured in gal (cm/s2). In Figure 2, the x axis are the different samples along the time. The y axis shows vertical, lateral, and sagittal acceleration values measured in gal.

To process these data, the standard deviation (σ) of 30,000 samples (in five minutes) is calculated. In this way the samples are adapted to the instants of time present on glucose measurements. The standard deviation from the past five minutes of acceleration data is associated with the glucose readings for that instant. The standard deviation of acceleration captures the intensity of the physical activity carried out in the past five minutes which will have an impact in upcoming glucose levels.

The standard deviation of the patient’s acceleration in the instants before taking the sample is calculated and the last five minutes of acceleration are taken. As the acceleration sensor takes 1 sample every 10 ms, 100 samples per second, and 6000 per minute are needed to do the five minutes. In total, 30,000 samples are required.

A 2-dimensional matrix is generated for acceleration data, the x axis being the different instants of time while the y axis captures the standard deviation of the past 30,000 samples measured in gal. The x axis are the different samples along the time and the y axis shows the standard deviation of 30,000 samples. The accelerometer provides acceleration data in three-dimensions, vertical, lateral, and sagittal. In Figure 3 the processed acceleration levels are shown.

These values are used for the training vector in the models that include the acceleration values. The two hour values (24 values every 5 min from t-23 to t) and the value after one hour (t + 12) are used. That group composes a sliding window and forms a prediction sample. In x, the 24 values from the past are used. An attempt is made to predict, with the predicted value being “y prediction” and the dataset value being “y test”. Two-hour values are used (24 values every 5 min from t-23 to t). That group composes a sliding window and forms a prediction sample.

In x, the 24 values from the past are used. Depending on the model, different values of acceleration, insulin, or food are added for training.

“y predicted” corresponds to the predicted value by the model (at t + 12). “y test” value corresponds to the dataset glucose values (for all t + 12 samples).

#### 3.1.3. Insulin Data Pre-Processing

Another input data signal used for glucose prediction is insulin. Type 1 diabetes patients need to take both fast and slow insulin doses in order to regulate BG levels all the time. In the dataset, these doses are injections of certain number of units of insulin. There are some models for insulin absorption by the human body [9]. In this paper we use some of these insulin absorption models imitating the body’s absorption of insulin in order to pre-process the insulin data and study their implications in the accuracy achieved by a recurrent neural network (RNN) based machine learning model.

In the corpus used, two types of insulin are used: rapid-acting and slow-acting insulin. The major characteristics for each type of insulin are:Rapid-acting insulin: Its effects begin between 5 and 15 min after your injection and have a peak of maximum effect of 1 to 2 h. Its action lasts between 4 and 6 h. This can vary a little depending on the dose of insulin administered and the specific type of insulin;Slow-acting insulin: Its effects begin between 1 and 2 h after the injection and does not have an absorption peak, but its action is relatively flat and lasts between 12 and 24 h depending on the specific type of insulin administered. It is usually used at night due to its long duration.

The main research question addressed in this manuscript is analyzing the influence that modeling the absorption curves for insulin and carbohydrates in the human body in order to pre-process the insulin and meal input data will help to improve the accuracy in BG predictions in a PH of 60 min for a RNN based machine learning model. Several absorption models for insulin and carbohydrates have been implemented. In particular, for insulin, three different absorption models depending on the type of insulin have been used in this paper. Firstly, a linear-segment based approximation model for fast-acting insulin described in [9] has been used. Secondly, the absorption profiles described in [21]. Thirdly, the insulin absorption has been approximated using two concatenated exponential segments. A first increasing exponential will model the process in which the rate of absorption of the units of insulin taken is higher that the rate of insulin elimination by the human body and will end at the peak (maximum value for the plasma insulin concentration). A second decreasing exponential will model the second part of the curve in which the insulin elimination from the human body is predominant.

#### 3.1.4. Haiya et al. Model

The model described in [9] is based on a set of differential equations that describe the different processes in the human body to absorb insulin and carbohydrates. In particular, the insulin-independent glucose uptake follows a Michaelis–Menten kinetics in which the rate of enzyme reactions ($V_{o}$) is based on the composition of the substrate, as shown in Equation (Equation 1).
(1)$$V_{o}=\frac{V_{max}\left[S\right]}{K_{m}+\left[S\right]}$$
where $V_{max}$ is the maximum reaction speed for that substrate, [S] is the substrate concentration and $K_{m}$ is the Michaelis constant.

To facilitate this absorption model, [9] proposes a temporal approximation for rapid insulin. This model of absorption for rapid (Lispro) insulin is given by the Equation (Equation 2).
(2)$$I_{t}=\left\{\begin{array}{cc} 0.25 & \mathrm{if}\;0\leq t\leq 5\\ 0.25+1(1+\frac{t-30}{30-5}) & \mathrm{if}\;5\leq t\leq 30\\ 0.25+1(1--\frac{t-30}{120-30}) & \mathrm{if}\;30\leq t\leq 120\\ 0.25 & \mathrm{if}\;120\leq t\leq 140 \end{array}\right.$$
where $I_{t}$ is the insulin absorption (mU/L) and *t* is the time in minutes.

Adapting Equation (Equation 2) to take into account the insulin dose, Equation (Equation 3) is obtained.
(3)$$I_{t}=\left\{\begin{array}{cc} 0.25 & \mathrm{if}\;0\leq t\leq 30\\ 0.25+1(1+\frac{t-120}{90}) & \mathrm{if}\;30\leq t\leq 120\\ 0.25+1(1--0.5\cdot \frac{t-120}{120}) & \mathrm{if}\;120\leq t\leq 240\\ 0.25+0.5(1--\frac{t-240}{240}) & \mathrm{if}\;240\leq t\leq 480 \end{array}\right.$$

By analyzing the models for Lisro and Regular, two representations of insulin uptake are generated over time in Figure 4a,b. Processing insulin over time produces a signal shown in Figure 4c.

#### 3.1.5. Insulin Absorption Profiles

In the Table 3, different types of insulin and the duration of absorption are displayed [22].

From the data in the Table 4, action curves of the different types of insulin shown in the Figure 5 have been generated [9]. Similar models for time-action profiles of insulin preparations are captured in [23].

Lispro insulin (in red) is very fast-acting. It starts working before the first 15 min (0.25 h). The peak is reached between the first and second hours, and lasts between four and six hours. Regular insulin (in blue) is fast-acting. It starts working after half or one hour. The peak is reached between the second and fourth hours, and lasts between six and eight hours. NPH insulin (in green) is intermediate-acting. It starts working in the first or second hour. The peak is reached between the sixth and tenth hours, and lasts more than 12 h.

In the dataset there is only Lispro (fast) and NPH (slow) intakes.

Firstly, these signals have been represented. Secondly, they have been multiplied by the units of insulin to take into account higher doses. Thirdly, the signal in Figure 5 is obtained.

#### 3.1.6. Insulin Absorption Exponentials

Another, simpler proposal is to take into account the data in Table 4 to generate exponential signals increasing until the insulin peak and decreasing until the effect of insulin absorption ends.

This method obtains signals in Figure 6a,b for both fast-acting and slow-acting insulin. It uses the data in Table 4 to perform increasing exponentials until the insulin peak and decreasing until the effect of insulin absorption ends.

In Figure 6a shows the fast insulin: the init is 15 min, the peak 120 min (2 h), and the duration is 300 min (5 h). In Figure 6b appear the slow insulin: the init is 60 min, the peak 360 min (6 h), and the duration is 720 min (12 h).

Then, these exponential are applied to the patient data and the signal in Figure 6c is obtained.

For this simulation, an exponential will be used that is increasing up to 15, 30, or 60 min depending on whether the food has less than 500 calories, between 500 and 900 or more than 900 calories, respectively, plus a decreasing exponential from the peak to completion. the 4 h of digestion. We will multiply the signal generated by the number of calories divided by 100 to have a higher peak when the calories ingested are higher.

#### 3.1.7. Food Data Pre-Processing

Food intake has a direct impact on future blood glucose levels. To achieve a good prediction for upcoming blood glucose levels, two major parameters have a particular relevance: the amount of carbohydrates ingested and the glycemic index (which indicates how fast the glucose reaches the bloodstream).

The dataset [18] used only contains information related to the amount of calories for some of the meals, annotated by a dietitian. The glycemic index was approximated using different absorption curves. Carbohydrate absorption curves have been modeled using the concatenation of an increasing and a decreasing exponential signals. In this way, we simulate how the ingested calories are absorbed over time and take into account how they affect the glucose level.

The rate at which carbohydrates are absorbed also depends on the chemical structure of the food. Some nutrients, such as cereals and derivatives, dairy products, fruits, legumes, and tubers, have a slower absorption while sugars, juices, soft drinks, or honey have faster absorption. In addition, the state of the food (solid or liquid) is also an influencing factor in the rate of absorption. As this information is not available in the dataset, a generic absorption curve in Figure 7a is used for all the meals. This generic curve is applied to patient data and the signal in Figure 7b is obtained.

#### 3.1.8. Windowing the Input Signals

The model proposed uses the information from meals, insulin (both fast and slow acting), physical activity, and glucose measurements, from a continuous glucose monitoring device in the last 2 h in order to estimate the blood glucose (BG) level in the prediction horizon of 1 h. The input signals are sampled every 5 min. The pre-processed signals for each T1DM patient are then windowed in order to feed the machine learning model that will provide an estimation for the BG level in the prediction horizon of 60 min.

Figure 8 shows the samples taken in a window of data and the position in time samples for the predicted value:

Each window of input data will consist of 24 samples (2 h from t-23 to t) for each of the 4 or 5 pre-processed input signals: past blood glucose levels, physical exercise, fast insulin, slow insulin, and carbohydrates. In the different experiments carried out to validate the proposed model, not all of the configurations had carbohydrates signal. This way, it is possible to analyze the importance of carbohydrates in predicting upcoming values for blood glucose levels. Additionally, the glucose level for the following hour (t + 12) is necessary in order to validate the prediction.

For each window of input data, a single output value is recorded in the inference. The output it is the blood glucose level after the prediction horizon (one hour).

The different input windows will be generated using a sliding approach. A single sample in time will be removed and the next sample in time added in order to generate a new window of data. All the night measurements were eliminated due to the lack of acceleration data during that time.

The total number of windows of data have been generated for each participating patient. From the 9 participants in the dataset [18], 6 patients have been selected because of the completeness of the available data. After pre-processing, there are a total of 4 or 5 input signals (blood glucose, fast acting insulin, slow acting insulin, carbohydrates, and physical activity) that provides samples every 5 min. These signals are windowed as shown in Table 4.

The patients with the highest number of windows are patients 001 and 008. It is expected to obtain better results with this patients.

### 3.2. LSTM Based RNN Proposed Architecture

To make the prediction of the patient’s blood glucose level in the prediction horizon (PH) of one hour after data acquisition, an LSTM long short term memory (LSTM) based recurrent neural network (RNN) model is proposed. A single LSTM layer is used as in previous studies, such as [3,24]. Deeper architectures, such as [25], have shown better prediction performance. However, the objective of the current manuscript is to compare how much a machine learning model can learn from pre-processed carbohydrate an insulin signals according to different absorption models and not to compare different machine learning methods.

A regression problem is solved using a RNN based model in order to assess the impact of different absorption curves for carbohydrates and insulin. A more complex (deeper) architecture for the machine learning method is likely to partially compensate a certain degree of inaccuracy in the absorption model. The model in [3] used a single layer bi-directional LSTM with 8 memory cells but added several dense layer after the bidirectional LSTM expanding the state to up to 64 hidden neurons. The model in [24] used a single LSTM layer with several configurations for the internal number of memory state cells, ranging from 8 to 128. A similar model in complexity will be used in this manuscript with a single LSTM layer of 56 memory units. Finally, a 1 unit output dense layer is added as the prediction of the blood glucose level is a single scalar value.

Figure 9 shows the architecture for the machine learning model used:

The total complexity of the model could be assessed by the number of trainable weights. There is a total of 13664 weights in the LSTM part of the model and 57 on the output dense layer. The input shape in order to feed the model is (None, 24, 4) or (None, 24, 5). Each sample for training or testing the model will be a bi-dimensional tensor with 24 time samples for 4 or 5 input signals. The first “None” parameter refers to a variable number of samples that can be fed to the model at the same time. The output shape of the LSTM is (None, 56) and the output shape of the dense layer is (None, 1), providing a single estimate for each input tensor.

The model has been implemented in Keras [26] as a sequential model and compiled by evaluating the loss function using the root mean square error (RMSE).

### 3.3. Train, Validation, and Test Separation

The model is fed with part of the windows of the input data for training (the training set) and the achieved score for the fidelity in the reconstruction for the predicted glucose level will be computed using a different set of input windows (test set). The training and test sets split is randomly done. The model is trained independently for each patient using 80% of the data. The remaining 20% is used for calculating the final score as testing data.

The training data are further divided into training and validation subsets that are taken into account for training the machine learning algorithm. From the training data, 75% of the windows are used for training and the remaining 25% for validating the convergence of the training process every iteration (epoch).

### 3.4. Evaluation Metric

The prediction evaluation is based on the losses obtained in the test samples when estimating the blood glucose level in a prediction horizon (PH) of 60 min. The loss function used is root mean square error (RMSE). Thus, the lower the error obtained, the better the predictions made by the model. The idea is that when prediction errors are small, the machine learning model is able to learn the internal patterns that transform the input signals into the output estimation.

### 3.5. Training the Model with 7 Configurations

A set of 7 experiments has been designed in order to pre-process the input data to train the machine learning model in Figure 9. Each experiment simulates a different absorption function for carbohydrates and insulin. The configurations for each experimental case are fully described in the next section.

The training process will run for a fixed number of iterations called epochs using a batch of instances to compute the gradients for numerical optimization. Every batch will provide an update in the weights of the model. In our case, the value of batch size is 16 and the number of epochs has been set to 150. These values have been selected in order to avoid over-fitting and under-fitting. In the case of over-fitting, the model would only adjust to particular cases and could not recognize new data. Additionally, in the case of under-fitting, the training would be insufficient and unable to make a good prediction.

## 4. Experiments Configuration Description

The blood glucose kinetics in the human body is a complex process that has been modeled in previous studies using a set of differential equations (Haiya et al. [9]). The intake of carbohydrates and insulin injections generate absorption processes that end up in having an influence in the future evolution of blood glucose levels. When trying to use a machine learning model to estimate future values for blood glucose concentrations, such as the model in Figure 9, the complex mechanisms in the human body should be learned by the model in order to transform the values for the input signals into prediction values which are as close as possible to the real ones. The objective of this manuscript is to assess the impact on the learning process by the machine learning model when providing additional information to the input data so that the amount of information that the model has to learn is minimized. In order to feed the machine learning model with input data which are closer to the physiological processes in the human body, the carbohydrate intake information and the insulin injection data will be converted into absorption curves using different absorption models. The amount of carbohydrates and insulin units will be convoluted with different models to approximate the meal digestion and insulin absorption by the human body. Different absorption curves have been used in order to generate different experiments. A description of each experiment is captured in this section while the particular results for each of them are described in the next section. In order to be able to compare these results in an impartial way, 10 different executions for each patient for each experiment set up have been carried out, using for each one of the executions, the same datasets (same time instants) for testing, training, and validation so that the results are comparable.

Table 5 summarizes the input signals used for each of the experiments and the absorption signal used for slow and fast insulin and to simulate food digestion. All the experiments use the raw glucose signal, as well as an estimation for how vigorous the physical activity is. The major differences are in the signals used to pre-process insulin and carbohydrates.

### 4.1. Experiment 1 Configuration

As captured in Table 5, the first experiment uses the raw samples for blood glucose, fast acting insulin, and slow acting insulin. The standard deviation of the acceleration data is added to this input to consider the physical activity of the patient. This first model does not have into account the information of the patient’s meals. The information about meals in the D1NAMO dataset [18] is based on the visual estimation for carbohydrates made by a nutritionist after analyzing the photos of the food taken by each participant. Some participants did not record all the meals. For some participants there were duplicated data. Finally, the exact time for the ingestion of the meals is not known and has been estimated by the time in which the photo was taken (assuming that it was take immediately before having the meal). For all of these reasons, we have left apart the information about meals in the first experiments and only used it in the last ones.

### 4.2. Experiment 2 Configuration

The second experiment also uses the glucose samples as an input. The standard deviation of the acceleration data is also added to this input to take into account the physical activity of the patient. Finally, the information for both fast and slow acting insulin is pre-processed using an exponential increasing and then decreasing absorption curve. The idea is that a simple absorption curve with and exponential decay will approximate the insulin absorption by the human body and, therefore, contribute to improve the estimation of the blood sugar levels that the patient will have one hour after the instant in which the prediction is made. In this experiment, as in the previous one, the information of the patient’s meals will not be taken into account.

### 4.3. Experiment 3 Configuration

The third experiment also uses the glucose signal and the standard deviation of the acceleration data as inputs, as in the previous experiments. The pre-processing for slow acting insulin uses the same simple absorption curve as in the previous experiment based on the concatenation of two exponential-shaped segments. The novelty of the experiment is for the pre-processing of the fast acting insulin which has been modeled following the proposal in Haiya et al. [9] and given by the Equation (Equation 2). The pre-processing model is expected to better describe the insulin absorption process in the human body and simplify the training for the machine learning model. In this experiment, as in the previous two, the information of the patient’s meals will not be taken into account.

### 4.4. Experiment 4 Configuration

The fourth experiment is based on experiment number 2 adding some basic information about meals. As in experiment number 2, the glucose samples are used together with the standard deviation of the acceleration data as a meter for the physical activity of the patient. As in experiment 2, the pre-processed fast and slow insulin are added as an increasing and then decreasing exponential depending on the type of insulin. The novelty for experiment number 4 is that we add the raw values for the meals as the estimated amount of calories taken in the instant of time in which the picture of the meal was taken. Carbohydrates play a crucial role on guiding the blood glucose kinetics and even with partial and not accurate estimations, the model is expected to be able to improve predictions.

### 4.5. Experiment 5 Configuration

The fifth experiment is based on experiment 4 but simulating the carbohydrate absorption using an exponential-shaped curve similar to the curve used for insulin absorption. The peak time is adjusted to simulate the digestion process. The past values for blood glucose levels, the standard deviation of the acceleration data and the pre-processed insulin following exponential-shaped absorption curves are used, as in experiment 4. The carbohydrate absorption curve will approximate the meal digestion process and we expect that the machine learning model will be able to achieve better prediction results.

### 4.6. Experiment 6 Configuration

The sixth experiment is based on experiment number 3 but adding the digestion approximation based on exponential-shaped absorption curves as in experiment number 5. The major difference with experiment number 5 is the use of the proposal in Haiya et al. [9] and given by the Equation (Equation 2) in order to approximate the fast acting insulin absorption.

### 4.7. Experiment 7 Configuration

The last experiment has used the activity profiles of different types of insulin in [22] in order to approximate the insulin absorption for both slow and fast acting insulin. The seventh experiment will use the glucose samples taken in the 24 moments prior to the prediction, corresponding to 2 h of patient monitoring and the standard deviation of the acceleration data as an indicator for physical activity, as in all the experiments. The information about meals will be pre-processed using the exponential-shaped curve. The major difference will be the use of absorption curves for insulin which will follow measured profiles according to the type of insulin.

## 5. Results

The results obtained using the machine learning model in Figure 9 based on the pre-processed data following the 7 different experiment set-ups as described in the previous section are captured in this section. The first subsection is dedicated to show the results per experiment for all the patients or participants. The second subsection is dedicated to show the results per patient or participant independently.

### 5.1. Results per Experiment

For each experiment, the data for the insulin injections and carbohydrate intakes from all the participants will be pre-processed using the absorption curves selected for each experiment, as described in the previous section. The model in Figure 9 will be trained for each participant 10 times (using different training sets) and average results will be calculated. Figure 10 captures some of the results for some of the data used for testing for all the 7 experiments. The figure shows the predictions for each of the experiment configurations (y_pred_exp) and the actual value in black (yTest). The model losses are evaluated with the root mean square error (RMSE) between the predicted result and the real measure (for the test set).

The results captured in this section will provide average values obtained as the average of the score obtained in 10 different executions to take into account the variations that occur according to which part of the data are used for training, validation, and testing. The following values have been used:RMSE for each execution. There are 10 executions for each patient and experiment number. In total, 420 individual scores have been computed. Each execution has a root mean square error per patient and experiment number;Mean, minimum, and maximum values for the 10 executions per patient. They are calculated from the 10 executions of each experiment number and patient. In total, 42 means, 42 minimums, and 42 maximums are computed.

Table 6 shows the results separated by experiment number. For each of the experiments carried out, the root mean square error (RMSE) obtained in the predictions for each execution for each participant is calculated. As 10 executions were carried out per participant to take into account possible variations, the mean of the RMSE values obtained in the 10 executions for all the participants is shown in the first column. The other columns show the results for the mean results for the maximum and minimum values per execution and the overall minimum and maximum values.

In particular, the meaning of the information captured in each of the columns in Table 6 is:Experiment number: to identify each configuration and the absorption curves used for insulin and carbohydrates;Mean of means: mean of all the patients mean values, by experiment number. There are 7 values, one per experiment number. The calculation per experiment is carried out in 2 steps. First, the mean of the results of the 10 executions per patient was found, and then the mean for the 6 patients was calculated;Mean of minimums: mean of the minimum values of the 10 executions of the same patient, by experiment number. First, the minimum of the 10 executions per patient results was found, and then the mean of the 6 patients was calculated;Minimum of minimums: minimum of the minimum of the 10 executions of the same patient, by experiment number. First, the minimum of the 10 executions was found, and then the minimum of the 6 patients was calculated;Mean of maximums: mean of the maximum values of the 10 executions of the same patient, by experiment number. First, the maximum of the 10 executions was found, and then the mean of the 6 patients was calculated;Maximum of maximums: maximum values of the maximum of the 10 executions of the same patient, by experiment number. First, the maximum of 10 executions was found, and then the maximum of the 6 patients was calculated.

RMSE values have been used in order to be able to compare results with previous studies which have published these values for their models. Another parameter that could be added in order to assess the accuracy of predictions is the Pearson’s correlation coefficient, such as in [27].

The ANOVA test for the RMSE values for the 7 different experiments shows statistically significant differences (*p*-values under 0.05). In particular, the *p*-value is $1.8510^{-13}$, F-value is 12.98, and critical F-value is 2.12.

In order to show the achieved results, a boxplot in Figure 11 is included. The results per experiment are captured. Each experiment aggregates the RMSE values in mmol/L for the 10 executions for the 6 participants. The best results are achieved for experiments 5 and 7 (both using absorption models for both insulin and carbohydrate information). The detailed results for each experiment configuration is described in a different sub-section.

#### 5.1.1. Experiment 1 Configuration

Experiment 1 is the base experiment in which no absorption curves will be used. It will constitute a similar scenario for the machine learning model in Figure 9, as previously used in related studies. The past values for the blood glucose signal will be combined with the accelerometer based activity estimation and the raw values for both fast acting and slow acting insulin injections. No food information will be used for this base experiment. The accuracy of this experiment will be used to assess how much the machine learning model is able to learn from the raw information in the dataset and how the results will be improved when adding the different absorption curves in the following experiments.

The accuracy results for each participant in the dataset for experiment 1 are captured in Figure 12. The data for each participant are divided into training (60%), validation (20%), and testing (20%) sets and the training is repeated 10 times with different data for each set. The mean value of each the patients’ mean is 0.863. The mean value of each patients’ minimum is 0.587. The mean value of each patients’ maximum is 1.268. The global minimum is 0.304 and the global maximum is 1.844.

Figure 12 shows different results for each participant. The accuracy of data for each participant could be assessed in Table 1 and it is likely to have an influence in the scores obtained per participant. Participants 1 and 8 are those with a bigger number of insulin injections recorder and are those achieving the best results after the training of the machine learning algorithm with raw data. Participants 4 and 7 are those achieving the worst results. Participant 4 has the smallest number of fast insulin injections and participant 7 has no information for slow acting insulin injections.

It shows that the best results are with patient 001 and the worst with patient 007.

#### 5.1.2. Experiment 2 Configuration

Experiment 2 adds a simple absorption model to the information in both the fast and slow acting insulin data. As described in the previous section, the model concatenates two exponential signal (the first one modeling the increasing part and the second one the decay after the peak value part). Regarding the 10 executions with experiment 2 configuration for each of the patients, the following results are obtained. The mean value of each the patients’ mean is 0.579. The mean value of each patients’ minimum is 0.460. The mean value of each patients’ maximum is 0.766. The global minimum is 0.251 and the global maximum is 1.095.

With the 10 executions with experiment 2 configuration, the boxplot in Figure 13 is created. The results significantly improve as visually compared with those in Figure 12. Figure 13 shows that the best results are with patient 001 and the worst with patient 004 which coincide with the patients with the biggest (total) and the smallest (fast acting) number of insulin injections available in the dataset.

#### 5.1.3. Experiment 3 Configuration

Experiment 3 improves the absorption model for the fast acting insulin according to the results in [9]. Regarding the 10 executions with experiment 3 configuration for each of the patients, the following results are obtained. The mean value of each the patients’ mean is 0.580. The mean value of each patients’ minimum is 0.472. The mean value of each patients’ maximum is 0.790. The global minimum is 0.267 and the global maximum is 1.220.

With the 10 executions with experiment 3 configuration, the boxplot in Figure 14 is created:

In this experiment the results are similar to those in the previous experiment showing that the exact shape of the absorption curve is not crucial for the achieved results (both curves had the same peak time and lasted the same amount of time).

#### 5.1.4. Experiment 4 Configuration

Experiment 4 has added the raw information about meals (which have not been taken into account in the previous experiments). Regarding the 10 executions with experiment 4 configuration for each of the patients, the following results are obtained. The mean value of each the patients’ mean is 0.588. The mean value of each patients’ minimum is 0.418. The mean value of each patients’ maximum is 0.805. The global minimum is 0.218 and the global maximum is 1.266.

With the 10 executions with experiment 4 configuration, the boxplot in Figure 15 is created:

Adding the raw data for meals is not able to improve the average results as compared with experiment 2. It shows that the best results are with patient 001 and the worst with patients 004 and 007 as in previous cases.

#### 5.1.5. Experiment 5 Configuration

Experiment 5 follows the same path previously taken in experiment 1 for insulin, now using an exponential shaped absorption curve for the carbohydrate intake pre-processing. The results improve as compared with the previous experiments (in fact, experiment 5 will provide the best average results). The details for the 10 executions with experiment 5 configuration for each of the patients are as follows. The mean value of each the patients’ mean is 0.510 (the best value in Table 6). The mean value of each patients’ minimum is 0.393. The mean value of each patients’ maximum is 0.657. The global minimum is 0.220 and the global maximum is 0.882.

With the 10 executions with experiment 5 configuration, the boxplot in Figure 16 is created.

The results in Figure 16 show that participants 4 and 7 improve the achieved scores while participant 8 is now the worst participant. Both participants 4 and 7 had a significant number of meals recorded and no duplicated data. The information for the meals recorded by participant 8 shows that there are 4 duplicated meals and, therefore, the carbohydrate estimation based on the information in the dataset is likely to contain errors.

#### 5.1.6. Experiment 6 Configuration

Experiment 6 uses the improved model for fast insulin absorption in experiment 3. The food absorption curve is not changed. Regarding the 10 executions with experiment 6 configuration for each of the patients, the following results are obtained. The mean value of each the patients’ mean is 0.565. The mean value of each patients’ minimum is 0.481. The mean value of each patients’ maximum is 0.678. The global minimum is 0.217 and the global maximum is 1.079.

With the 10 executions with experiment 6 configuration, the boxplot in Figure 17 is created:

Again, experiment 6 is not able to improve the results in experiment 5 which is consistent with previous results.

#### 5.1.7. Experiment 7 Configuration

The final experiment uses insulin absorption curves as previously measured for real cases in order to further improve the absorption model for insulin (both fast and slow acting insulin types). The shape of the meal absorption curve is kept the same as in the previous experiment. Regarding the 10 executions with experiment 7 configuration for each of the patients, the following results are obtained. The mean value of each the patients’ mean is 0.515. The mean value of each patients’ minimum is 0.394. The mean value of each patients’ maximum is 0.709. The global minimum is 0.235 and the global maximum is 1.005.

With the 10 executions with experiment 7 configuration, the boxplot in Figure 18 is created:

The results are similar to those achieved for experiment 5 showing the best values for all the experiments.

#### 5.1.8. Results per Experiment Final Analysis

Table 6 shows the mean value, mean of minimums, global minimum, mean of maximums, and global maximum of the 60 executions (10 by each patient) for each experiment. Regarding the mean, the best results are for experiments 5 (0.510 mmol/L) and 7 (0.515 mmol/L). Those experiments used an exponential-shaped absorption curve for the carbohydrate information and a similar shape for fast acting insulin (experiment 5) or a real (measured) absorption curve for both the fast and slow acting insulin types. Experiment 5 also provides the best results in the mean of minimums (0.393), mean of maximum (0.657), and global maximum (0.882). All the absorption models significantly improved the results obtained for the base configuration using raw data (experiment 1) showing that a machine learning model will better learn glucose patterns from signals which mimic the insulin and carbohydrate kinetic processes in the human body.

In Figure 19, the overall boxplots by experiment number are shown in order to facilitate the visual comparison of the results achieved in all of them. The differences by patient number for each experiment configuration can be interpreted.

The obtained results indicate that experiment 1 (Figure 19a) (which has been normally used in previous research studies) does not provide good enough results. In experiment 2 (Figure 19b) the results improve, which reflects the difference between experiment 1 (which only uses raw insulin data) and simulates the insulin assimilation process with exponentials to make the glucose prediction. Experiment 3 (Figure 19c) does not show a significant improvement in the results when using the model in [9], validating that exponential-shaped absorption curves work well for the majority of the participants. Experiment 4 (Figure 19d) uses input data, which is similar to experiment 2, but adds the raw information for meals (without using digestion curves). The error achieved is very similar to the one in previous experiments, so the raw data of the food does not provide much information for the prediction (at least the machine learning model is not able to learn from it). Experiments 5, 6, and 7 (Figure 19e–g) provide acceptable results. In these cases, both a model for insulin absorption and another model for carbohydrate absorption are used instead of using raw data. The fifth experiment, which uses glucose, acceleration, insulin processed as exponentials, and the carbohydrate absorption model is the one that provides the most accurate predictions. Comparing all the experiment numbers, the results show that the lower RMSE values are for experiments 5 and 7 for most of the patients. In most of the patients, experiment 5 and 7 are the best, except with patient 008 (Figure 19g), where there are duplicates in the food data.

As a conclusion, the RNN based machine learning model is not able to learn well enough from raw data as compared to using absorption curves that mimic the physiological insulin and carbohydrates absorption processes in the human body. The results are, therefore, improved by pre-processing the signals with a linear model, a generic body assimilation curve or using exponentials. Comparing the models with or without carbohydrates, it can be seen that food processed data clearly enhances the performance of the prediction.

#### 5.1.9. Comparing Results with Those When No Activity Data Are Used

Activity data have been used in previous models for blood glucose estimations, such as [28]. Accelerometer sensors are able to capture movement related data. Previous studies have estimated the consumed energy based on acceleration data [28] and use several minutes of acceleration as a feature used to estimate future values of blood glucose levels [28]. We have used a similar computation based on the standard deviation of acceleration data over the last 5 min. In order to compare the results with those when not using the acceleration data, we have used the base model in experiment 1 without using the activity data as an input. Table 7 shows the results. The mean values for the RMSE errors are higher when the activity data are not used. The same happens with the minimum and maximum values.

Figure 20 captures the results for experiment 1 with and without using the activity data per patient. There is a significant improvement in all participants except for patient number 007 which provides similar results in both cases.

### 5.2. Results According to the Patient Number

In Figure 21 the boxplots by patient are shown and the differences by the experiment configuration are possible to be studied.

As represented in Table 4, the number of training windows differs among patients (depending on the amount of data per patient in the dataset). The patients with the highest number of training windows are patients 001 (663), and 008 (514). Patient 008 presented duplicates in the meals and the results achieved for this participant were worse than those for other patients even if a bigger number of windows of data could have helped the training of the machine learning model. In configuration 1, 2, and 3 (Figure 19a–c), the best results are from patients 001 and 006. In configuration 4, 5, 6, and 7 (Figure 19d–g) the best results are from patients 001, 002, and 006. The best results are obtained with patient 001 and the worst with patients 004, 007, and 008, due to the incompleteness of some of the meals that were not recorded and duplicates in the dataset.

In patient 001 (Figure 21b), the results are good in all cases except for experiment configuration 1 (using raw insulin data). The information in the dataset for patient 1 is the most complete one both for the meals and insulin and, therefore, the results when adding absorption curves provide optimal results. In patient 002 (Figure 21c), the results are good too and the best configurations are 3, 5, and 7. In patient 004 (Figure 21d), there is a significant improvement in experiment 7 (insulin profiles) with the presence of outliers and experiment 5 (insulin exponentials). In patient 006 (Figure 21e), the best results are for experiments 2 (no food signal) and 4 (raw signal). In this case, the food exponential is not improving the results. Patient 6 only provides information for 7 of the meals (in a 4 days period) and, therefore, some of the meals are likely missing in the input data. Using food intake information is not able to improve results. In patient 007 (Figure 21f), the results are not good, but the best results are 3 and 5. Participant 007 had no information for slow acting insulin in the dataset. In patient 008 (Figure 21g), the best results are from experiment 3. The information in the meals dataset for participant 008 show 4 duplicate meals and, therefore, the results do not behave well when adding a meal absorption curve.

## 6. Comparison with Similar Previous Studies

Absorption models for carbohydrate and insulin which mimic the human body absorption processes have been proposed in the previous sections, showing that for the D1NAMO dataset [18] the accuracy achieved by a machine learning model (shown in Figure 9) when estimating future values for blood glucose levels with a 60 min prediction horizon significantly improves as compared with the base case in which raw data for food and insulin are used. In this section, a comparison with previous studies using a similar machine learning model when applied to raw data in different datasets in order to obtain predictions for future values for blood glucose levels will be presented in order to generalize results to different datasets. In order to make a fair comparison between different previous studies, some characteristics are required:Model input: datasets containing CGM data for real participants, not simulated patients. Simulators normally capture simplified models with less complex patterns and machine learning models tend to provide better results for simulated data;Prediction horizon (PH): 60 min. Previous studies have used 30 and 60 min prediction horizons. The bigger the PH the less accurate the estimations will be;Method used: recurrent neural network–long short term memory. RNN with LSTM cells is the machine learning model selected in the current manuscript and has been successfully used in previous studies. In order to have a fair comparison of results among different studies, the complexity of the models measured as the number of trainable parameters should be similar. The model used in this paper uses a single LSTM layer with 64 memory cells and an output dense layer with a single neuron. Using more complex models with more layers will be able to learn more complex patterns from the data providing better results. However, the objective of the current manuscript is not to optimize the machine learning model but to compare how a single model will be able to learn the different patterns from data using different absorption curves;Pre-processing: in order to study the effect of insulin and carbohydrate absorption curves, this paper has defined and measured the results for experiment 1 in which raw data were used to train the model (no absorption curves were used for experiment 1). Previous papers with no pre-processing are going to be compared with experiment 1 in this manuscript;Output error values: RMSE for blood glucose level predictions are used. In the present work the values are in mmol/L. However, the values are converted into mg/dL because this unit is used more often in previous studies.

Table 8 summarizes the comparison with similar previous studies. Sun et al. [3] and Martisson et al. [24] used a similar RNN–LSTM one layer machine learning model applied to data from real participants. The prediction horizon was 60 min in both cases. No pre-processing for the insulin and carbohydrate signals was used. The results show worse figures for BG prediction values (RMSEs of 36.918 and 33.2 (mg/dL)) as compared to the results obtained in this paper for experiment 1.

In this work, the RMSE values have been computed with glucose levels in mmol/L. However, it is important to convert the units to mg/dL. In case of the first experiment, the result is 0.863 (mmol/L) = 15.5 (mg/dL). The result is improved in the fifth experiment with a mean value of 0.510 (mmol/L) = 9.2 (mg/dL).

De Bois et al. [2] found better results using a similar machine learning model than the base experiment 1 in this paper. They achieved an RMSE value of 7.08 mg/dL. The model architecture is quite similar. However, the data were for simulated patients and the prediction horizon considered was 30 min. Hence, their research work faced a more favorable case which is, therefore, not fully comparable to the results in this paper.

Munoz-Organero [25] also obtained better results than the base experiment in this paper. In this case, the study used the same dataset and prediction horizon getting results of 6.42 mg/dL. However, the research study in [25] used a more complex machine learning model which included pre-processing stages to simulate the carbohydrate and insulin absorption by the human body. Using absorption curves, as proposed in this paper, the intuition is that the input data to the machine learning model reduces the complexity of the patterns to be learnt as compared to models, such as [25], which propose deeper architectures that try to learn absorption curves based on raw data. Deeper models require more data to be trained which in some cases could require T1DM patients to have to train the system for a longer period of time before being able to use it.

Data in Table 8 show that a simple machine learning model can better learn from data if absorption curves for insulin and carbohydrate are used and achieve results similar to those of a more complex model. As a future work we will apply absorption curves to more complex models in order to optimize prediction results. Different absorption curves in insulin and food carbohydrates have been studied showing the improvement of results.

## 7. Discussion

Diabetes is one of the most frequent chronic diseases that does not have a cure and an exact treatment. Additionally, there are some complications attached, such as dementia, heart attacks, chronic dermatitis, and lack of blood supply. Diabetic patients should avoid hypoglycemia and hyperglycemia episodes. Machine learning predictive models have shown to provide good results for estimating future values for BG levels. Using these estimations, T1DM patients are able to adjust insulin and carbohydrate intakes and prevent hypoglycemia and hyperglycemia episodes. Moreover, the quality of life of patients could be improved and healthcare costs reduced. That is possible by predicting accurate blood glucose levels in the near future in diabetic type I patients.

The results in this manuscript show that machine learning models for predicting upcoming values for BG levels can better learn if absorption curves are used in order to pre-process insulin and carbohydrate raw data. A PH of 60 min has been used.

In this work, data from 9 patients suffering type 1 diabetes have been considered. Some control variables are physical exercise, food intake, injected insulin, and the body’s absorption processes. It has been necessary to perform a pre-experiment data quality assessment because inaccurate input values can negatively affect the results for the machine learning algorithm. Out of the 9 patients, 6 participants have been finally selected removing those that were suspected to have important information missing or duplicates.

Four ways of insulin absorption modeling have been used. First of all, no absorption model has been studied in order to have a base scenario and to facilitate the comparison with previous studies. Secondly, a linear model described in [9] has been used. Thirdly, assimilation curves directly provided by the insulin manufacturer have been studied. Lastly, a new model with exponential curves was developed for this work.

Additionally, there are two types of insulin used in the dataset: fast acting which lasts between 4 and 6 h and slow acting, which lasts between 12 and 24 h.

The machine learning model used is based on RNN with LSTM cells which has been widely used in previous studies, such as Sun et al. [3] and Martisson et al. [24]. For evaluating the performance of the models, a root mean square error (RMSE) based loss function has been used. The smaller the error found in the test data, the better for the model and the predictions.

Results show that using the exponential model the predictions of the blood glucose levels in the next hour improve for the majority of patients. To verify this, different tests were carried out modifying the absorption models and carrying out 10 different executions in each test.

The achieved results are encouraging, obtaining accuracy levels around 0.510 mmol/L (9.2 mg/dl) in the best scenario.

The results are promising, with low errors in the prediction horizon even for a simple machine learning model. The conclusion is that a simple model with good pre-processing of signals for real patient data can improve the performance of the prediction.

Some applications of this work in diabetes are improvements in blood glucose control, recommendations, nutritional advice, individualized treatment, as well as blood glucose level regulation when monitoring night-time insulin values with an insulin pump.

Future improvements include designing an automatic control system. For example, an artificial pancreas that controls the amount of insulin that is injected, to be able to do it automatically at times when the patient cannot. When controlling the amount, the glucose prediction values for the near future are used, since there is a relationship between glucose levels and the amount of insulin needed.

Regarding future work, we expect to improve the model, optimizing the hyperparameters and modifying the number of layers or memory cells.

## Figures and Tables

**Figure 1 sensors-21-05273-f001:**
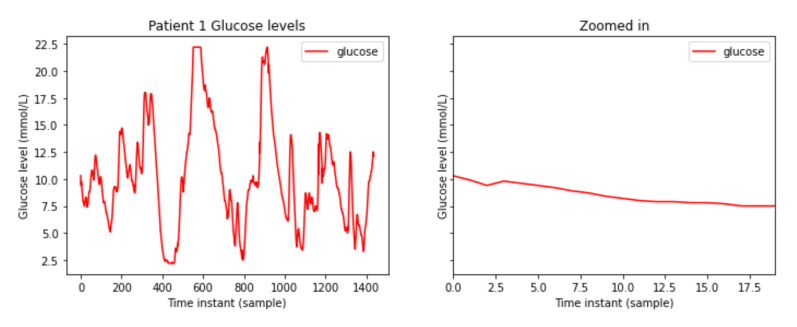
Patient 1 glucose blood levels.

**Figure 2 sensors-21-05273-f002:**
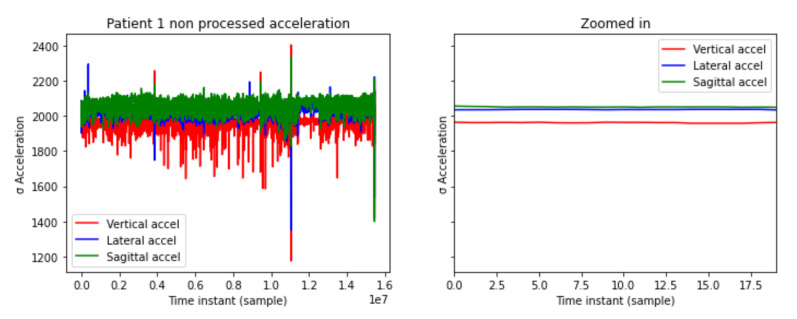
Patient 1 acceleration levels.

**Figure 3 sensors-21-05273-f003:**
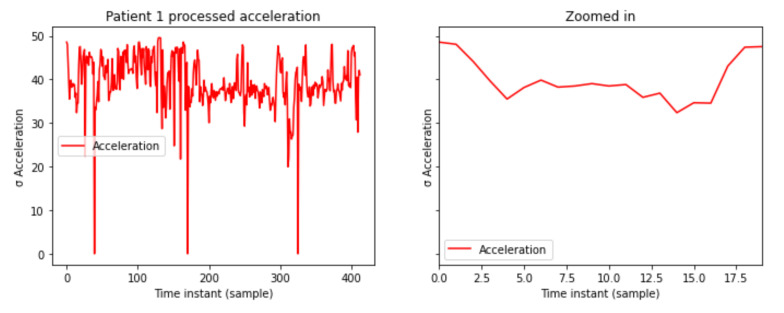
Patient 1 processed acceleration levels.

**Figure 4 sensors-21-05273-f004:**
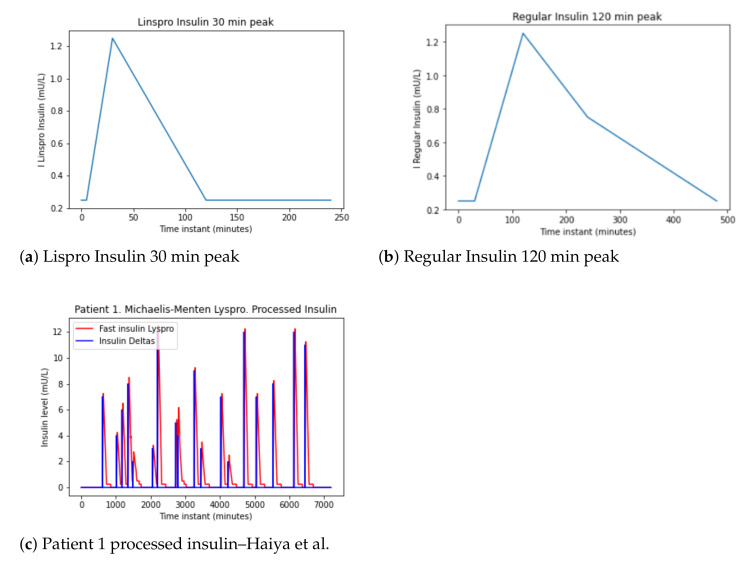
Patient 1 insulin data pre-processing—Haiya et al.

**Figure 5 sensors-21-05273-f005:**
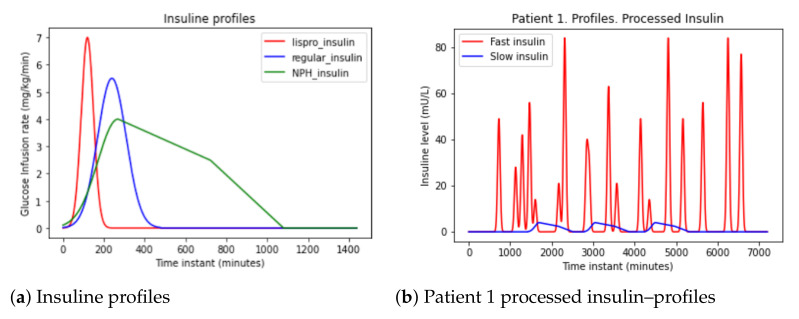
Patient 1 insulin data pre-processing-profile.

**Figure 6 sensors-21-05273-f006:**
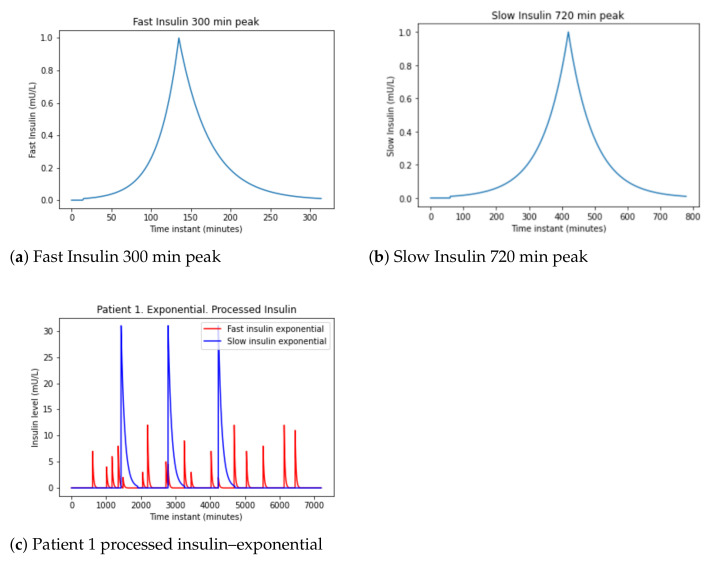
Patient 1 insulin data pre-processing-exponential.

**Figure 7 sensors-21-05273-f007:**
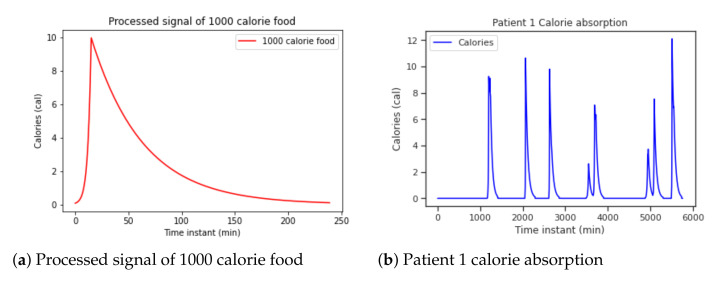
Patient 1 food data pre-processing.

**Figure 8 sensors-21-05273-f008:**
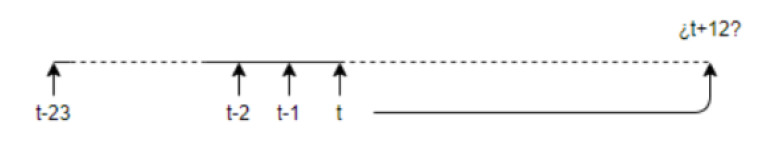
Windows diagram.

**Figure 9 sensors-21-05273-f009:**
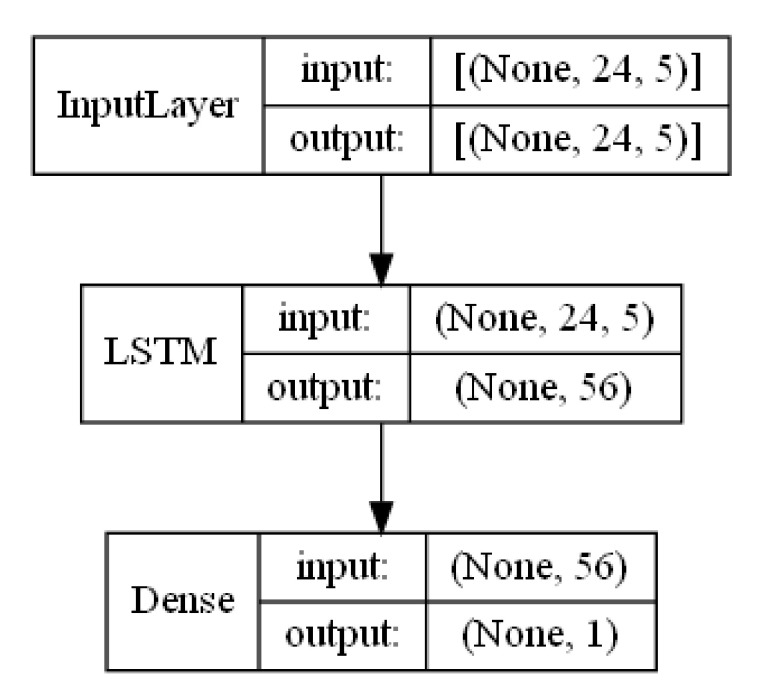
Model architecture with LSTM and Dense Layer.

**Figure 10 sensors-21-05273-f010:**
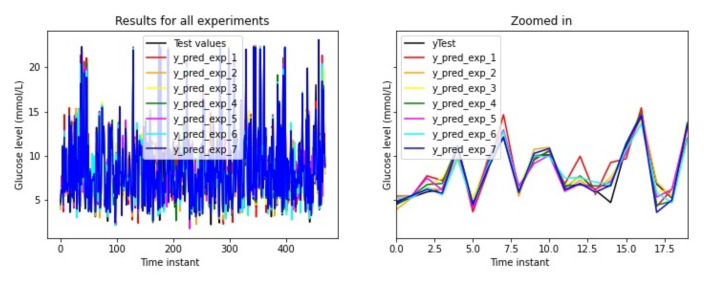
Experiment 1 boxplot scores, from 10 executions for each patient and configuration.

**Figure 11 sensors-21-05273-f011:**
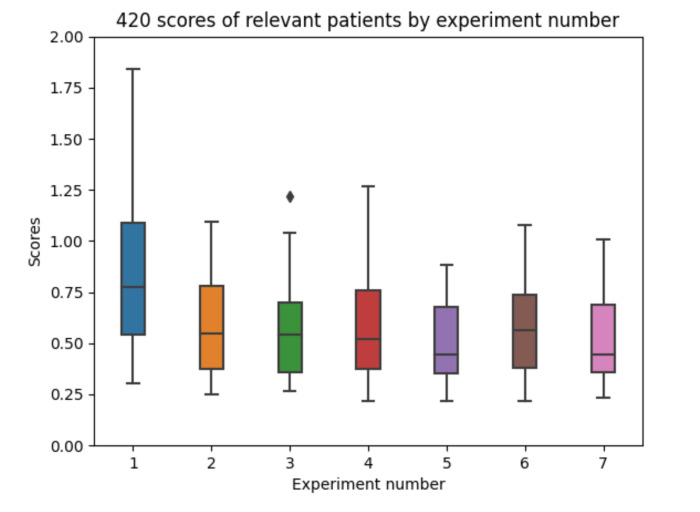
Experiments boxplot scores, from 10 executions for each patient and configuration.

**Figure 12 sensors-21-05273-f012:**
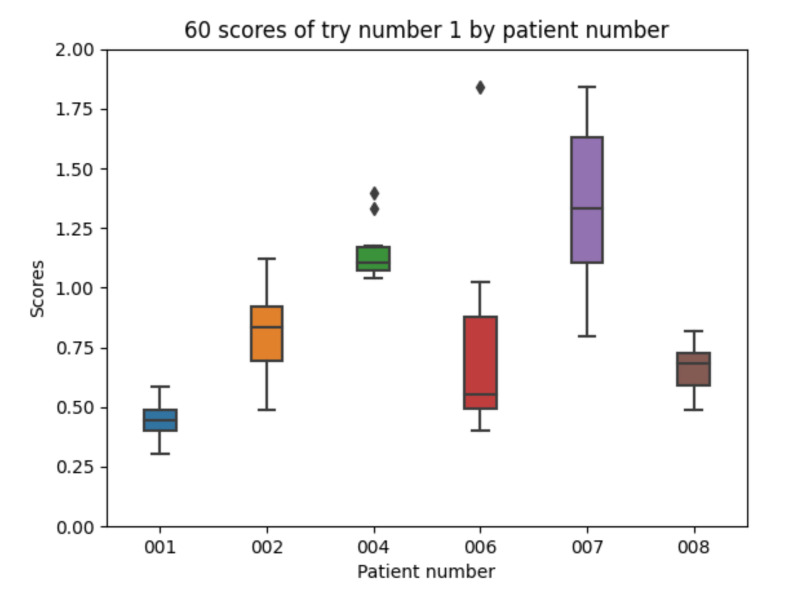
Experiment 1 boxplot scores, from 10 executions for each patient and configuration.

**Figure 13 sensors-21-05273-f013:**
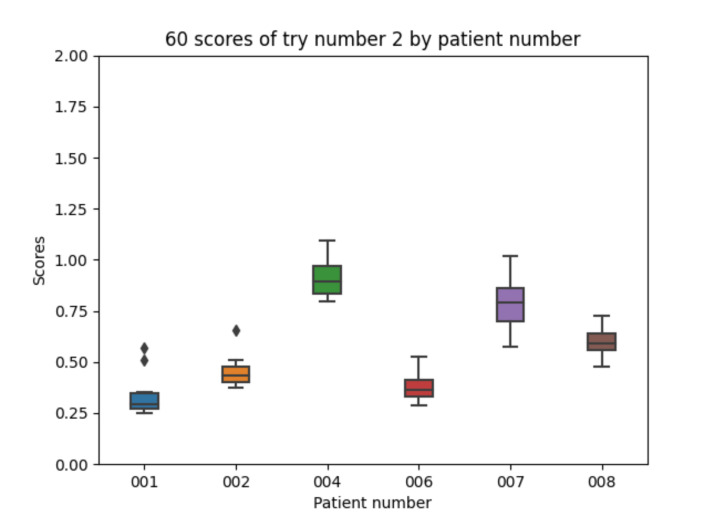
Experiment 2 boxplot scores, from 10 executions for each patient and configuration.

**Figure 14 sensors-21-05273-f014:**
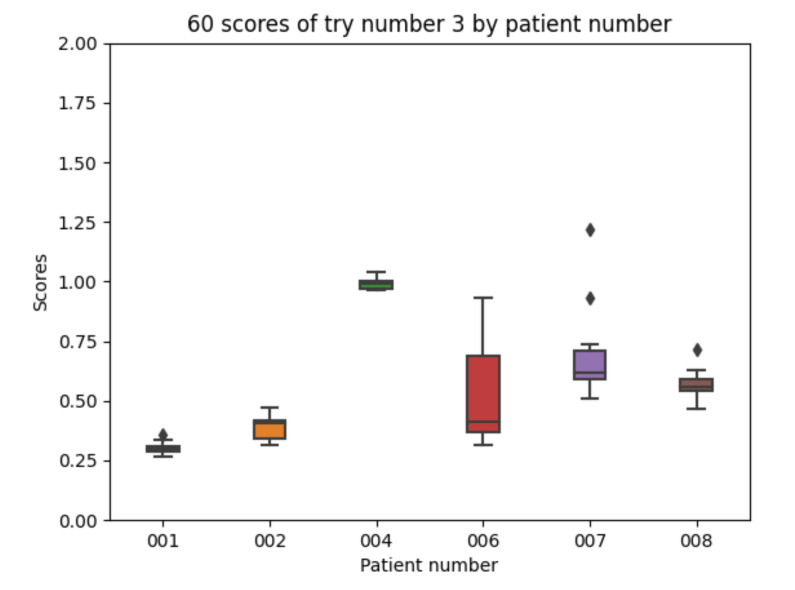
Experiment 3 boxplot scores, from 10 executions for each patient and configuration.

**Figure 15 sensors-21-05273-f015:**
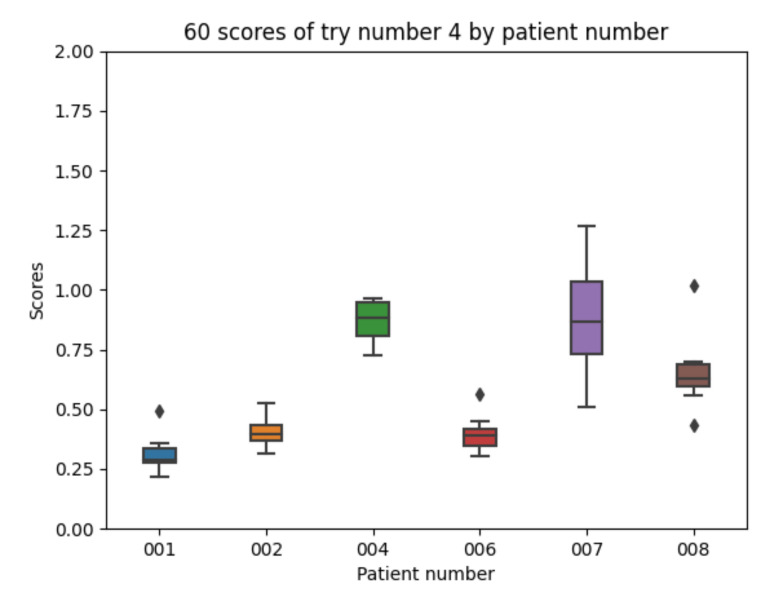
Experiment 4 boxplot scores, from 10 executions for each patient and configuration.

**Figure 16 sensors-21-05273-f016:**
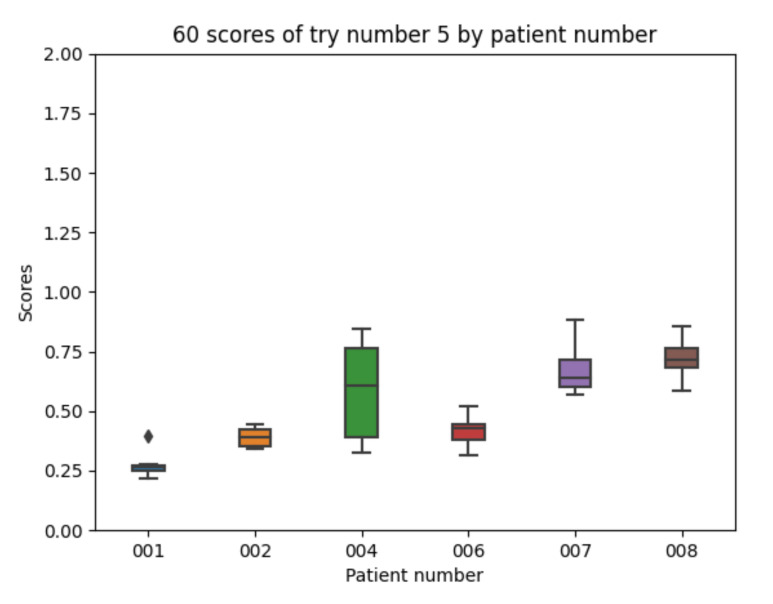
Experiment 5 boxplot scores, from 10 executions for each patient and configuration.

**Figure 17 sensors-21-05273-f017:**
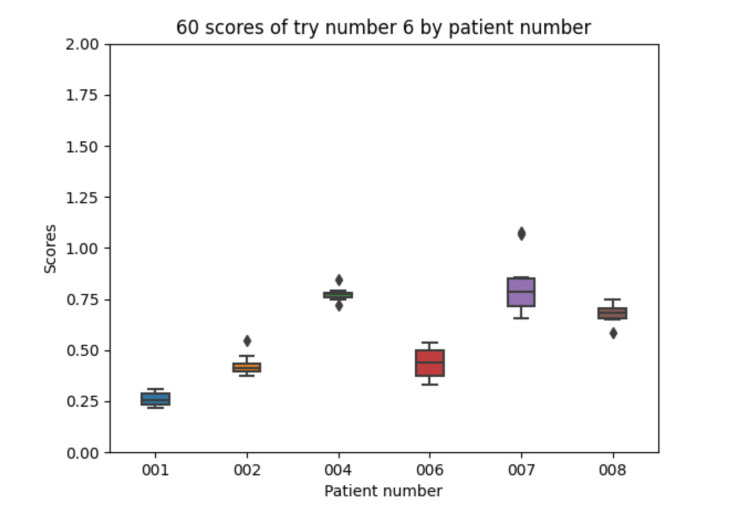
Experiment 6 boxplot scores, from 10 executions for each patient and configuration.

**Figure 18 sensors-21-05273-f018:**
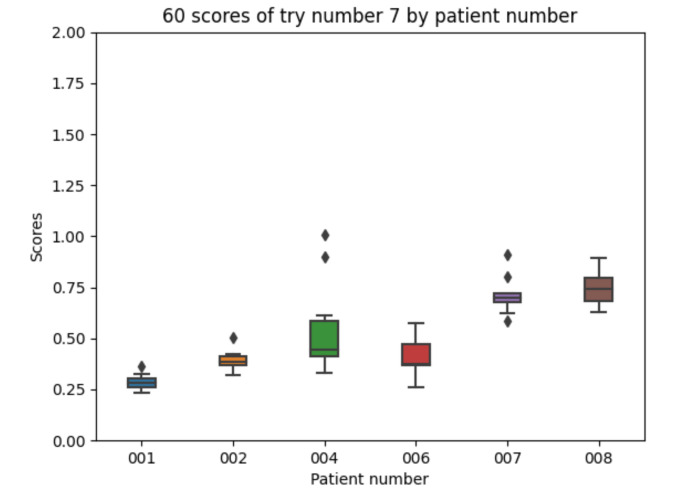
Experiment 7 boxplot scores, from 10 executions for each patient and configuration.

**Figure 19 sensors-21-05273-f019:**
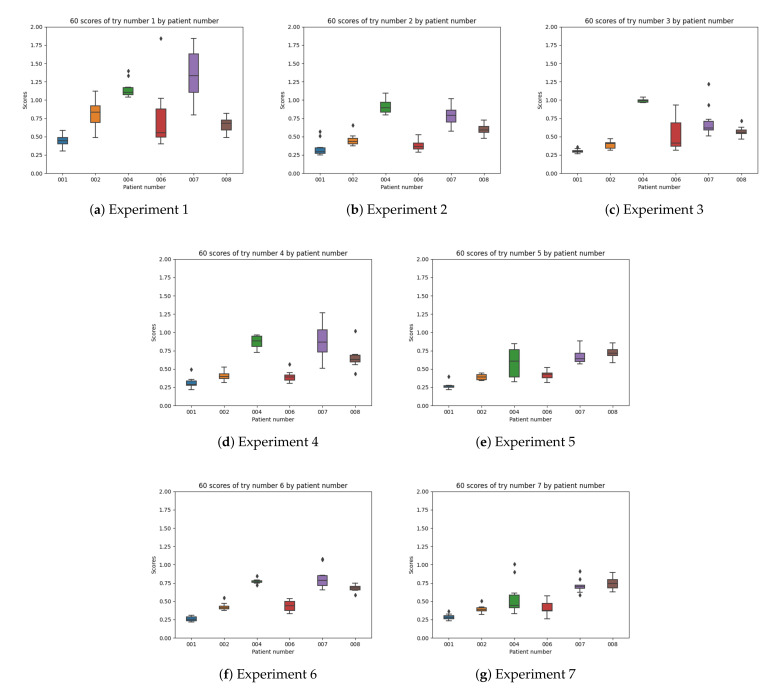
Experiments boxplot.

**Figure 20 sensors-21-05273-f020:**
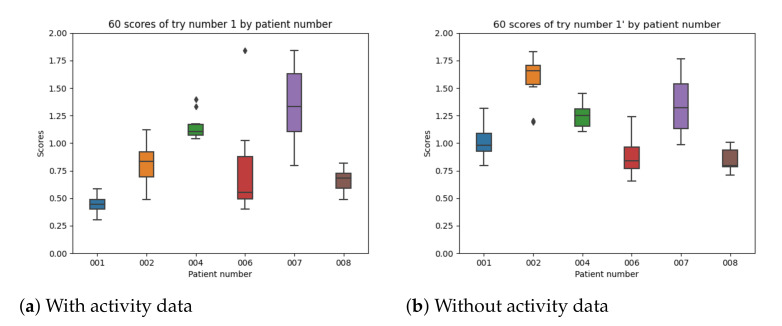
Experiment 1 boxplot with and without activity data.

**Figure 21 sensors-21-05273-f021:**
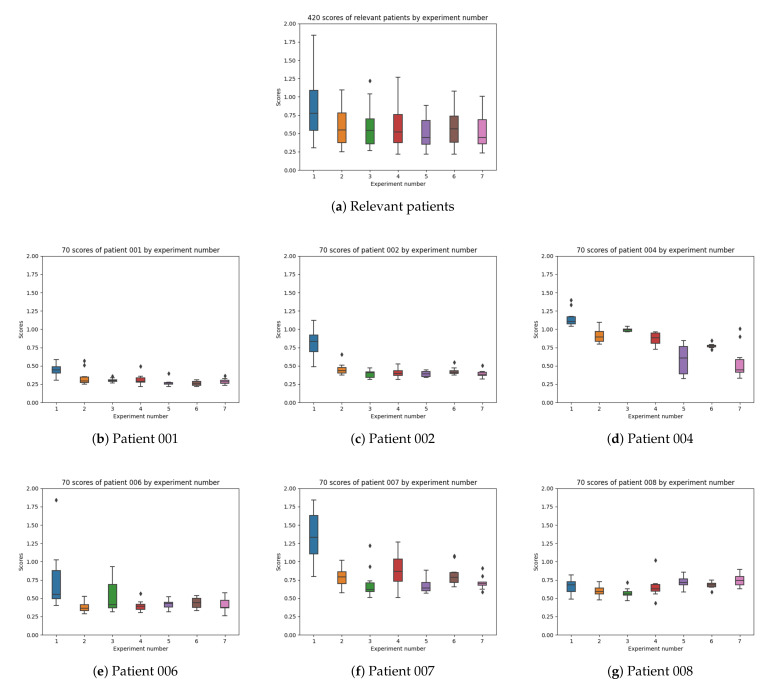
Patients boxplot.

**Table 1 sensors-21-05273-t001:** Information about T1DM patients in the D1NAMO dataset.

Patient Number	Number of Meals	Meal Incidences	Number of Fast Acting Insulin Injections	Number of Slow Acting Insulin Injections
001	16	1 meal repeated	19	3
002	8	–	11	2
003	3	–	3	0
004	7	–	10	4
005	9	no dates available	7	0
006	7	–	10	0
007	20	–	16	0
008	12	4 meals repeated	22	0
009	8	1 meal repeated	12	4

**Table 2 sensors-21-05273-t002:** Extended information about relevant T1DM patients in the D1NAMO dataset.

Patient Number	Year Range	Gender	Height Range (cm)	Weight Range (kg)
001	NA	Man	180–189	80–89
002	20–29	Man	170–179	60–69
004	20–29	Man	180–189	80–89
006	30–39	Man	190–199	70–79
007	30–39	Woman	160–169	70–79
008	60–69	Woman	150–159	50–59

**Table 3 sensors-21-05273-t003:** Types of insulin and characteristics.

Type of Insulin	Onset(h)	Peak (h)	Total Duration (h)
Fast-acting:			
Regular/normal	0.5–1	2–4	6–8
Lyspro/Aspart/Glulisine	<0.25	1–2	4–6
Intermediate-acting:			
NPH	1–2	6–10	<12

**Table 4 sensors-21-05273-t004:** Number of windows and total hours per patient.

Patient Number	Total Windows	Train Windows	Validation Windows	Test Windows	Total Hours
001	665	399	133	133	66:15:01
002	362	216	73	73	72:37:02
004	358	214	72	72	80:34:58
006	361	216	72	73	71:55:08
007	480	288	96	96	82:10:01
008	514	308	103	103	82:10:01

**Table 5 sensors-21-05273-t005:** Tries with different model configurations.

Experiment Number	Glucose	Acceleration	Fast Insulin	Slow Insulin	Food
1	x	x	Raw data	Raw data	
2	x	x	Exponential	Exponential	
3	x	x	Haiya et al. [9]	Exponential	
4	x	x	Exponential	Exponential	Raw data
5	x	x	Exponential	Exponential	Exponential
6	x	x	Haiya et al. [9]	Exponential	Exponential
7	x	x	Profiles	Profiles	Exponential

**Table 6 sensors-21-05273-t006:** Obtained results of each experiment number in mmol/L.

Experiment Number	RMSE Mean (mean)	RMSE Mean (min)	RMSE Min (min)	RMSE Mean (max)	RMSE Max (max)
1	0.863	0.587	0.304	1.268	1.844
2	0.579	0.460	0.251	0.766	1.095
3	0.580	0.472	0.267	0.790	1.220
4	0.588	0.418	0.218	0.805	1.266
5	**0.510** ${}^{1}$	**0.393**	0.220	**0.657**	**0.882**
6	0.565	0.481	**0.217**	0.678	1.079
7	0.515	0.394	0.235	0.709	1.005

${}^{1}$ Best values in bold format. Second best values in underline format.

**Table 7 sensors-21-05273-t007:** Obtained results for experiment 1 with and without activity data in mmol/L.

Experiment Number	RMSE Mean (mean)	RMSE Mean (min)	RMSE Min (min)	RMSE Mean (max)	RMSE Max (max)
1 (with activity data)	**0.863** ${}^{1}$	**0.587**	**0.304**	**1.268**	**1.844**
1 (without activity data)	1.159	0.907	0.657	1.500	2.226

${}^{1}$ Best values in bold format.

**Table 8 sensors-21-05273-t008:** Results of glucose predictions in a comparative study. Source: “Deep Physiological Model for Blood Glucose Prediction in T1DM Patients” (2020).

Study	Input Value	Prediction Horizon	Method Used	Pre-processing	RMSE
Sun et al. [3]	CGM data (Real and simulated)	60 min	RNN–LSTM + BiLSTM (1 layer)	-	36.918 (mg/dL)
Martinsson et al. [24]	CGM data (Real)	60 min	RNN–LSTM	Missing data	33.2 (mg/dL)
**Experiment 1 (raw data) This paper**	CGM data, insulin and carbohydrate (D1NAMO)	60 min	RNN–LSTM (1 layer, 64 units)	Exponential of insulin and food intake	15.5 (mg/dL)
**Experiment 5 (exponentials) This paper**	CGM data, insulin and carbohydrate (D1NAMO)	60 min	RNN–LSTM (1 layer, 64 units)	Exponential of linsulin and food intake	9.2 (mg/dL)
De Bois et al. [2]	CGM data, insulin and carbohydrate (simulated diabetic metabolic simulator)	30 min (With 60 min data history)	RNN–LSTM (1 layer, 64 units)	Data rearrangement, 3 way data splitting, standardization	7.08 (mg/dL)
Munoz-Organero [25]	CGM data, insulin and carbohydrate (Real) D1NAMO	60 min	Metabolic inspired model using RNN–LSTM (10 unit cells)	Signal fitting	6.42 (mg/dL)

## Data Availability

D1NAMO dataset: https://www.sciencedirect.com/science/article/pii/S2352914818301059 (accessed on 4 June 2021).

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
