# Peer review of "Using Absorption Models for Insulin and Carbohydrates and Deep Leaning to Improve Glucose Level Predictions"

_sensors, 2021, doi:10.3390/s21165273_

Round 1

Reviewer 1 Report

The paper presents a method for short term blood glucose level prediction that is validated on a public dataset. The core idea is that with data 'preprocessing', the accuracy is expected to improve. The idea is good, however, there are two problems that require major improvements before the paper can be considered for publication.

1. methodological issues
The authors put the emphasis on the effect of absorption profiles on the accuracy, but the data set does not really support this kind of research, because there is no information about food composition--note that not only the glycemic index is missing, the fiber and fat content also modifies the shape of the absorption curve. Experimenting with various absorption curve shapes for all patients and for all meals is a slight methodological error (no wonder it did not result in improvement) because the shape or the curve is determined by the meal much more than by the patient. Similarly, for the insulin profiles, the manufacturer data sheet should be used. In my opinion, this dataset would better support a study on the effect of physical activity on BGL prediction accuracy due to the good quality acc. data, i.e. training a model with and without this information. 
Another problem is that with 7 setups the focus of the paper is blurred, the reader is expected to be bored. The authors should select 2 or maximum 3 setups by dropping the trivial configs (e.g. those without food info). Selected models should be better analysed (e.g. check the effect of training data selection by cross validation). 
I also miss the discussion of the practical applicability of the method. The authors should mention that the method requires a CGM to be worn at all times, therefore the results are applicable only closed loop (CGM-insulin pump) setups.

2. style and format of the text
The paper is too long and overly redundant. I estimate that half of the current volume would be enough for this content. Some points are repeated 3 or more times. Examples are sect. 2.4 and the lines 404-416, 331-336, 241-246. This will bore the readers.
Also, the style is superfluous and over-complicated over the whole text, e.g. lines 376-377 can be simply phrased as better accuracy. The style should be more formal cf. line 205 '...needed to do the five minutes'

Minor remarks

The nights should be separated from the data because of the lack of movement and glucose infusion (the modeling challenge is quite different).

The main conclusions about the differences in accuracies should be statistically proven. The term 'significant' should not be used without such checks. Similary, the term 'optimized' would involve that the authors can prove there is no better solution (use 'improved' instead, also in the title).

The lines 50-80 should separate the absorption model from the BGL regulation model.

In lines 90-120, focus on studies that are directly comparable to this study.

Table 1 should include only the 6 patients actually used, and be extended with information on gender, age, HgA1c, number of days and CGM records and lifestyle (inpatient or outpatient).

The accel. data is very valuable. However, it should be proven by argument or references that the stdev feature correctly depicts the volume/intensity of the activity.

Include only information relevant to the study. In table 2, only Lispro and NPH should appear.

The heurisic profiles used for glucose infusion do not resemble the curves of absorption models (the peak is not so pointed and sharp).

Table 3 should include the number of validation windows as well.

Table captions should provide a full explanation, e.g. in table 5, bold and underline should be explained.

Author Response

All the comments from the reviewers have been fully considered and implemented in order to improve the current version of the manuscript. We want to thank the reviewers for their feedback and comments that have allowed us to improve the quality of the paper.

Reviewer 1.

The paper presents a method for short term blood glucose level prediction that is validated on a public dataset. The core idea is that with data 'preprocessing', the accuracy is expected to improve. The idea is good, however, there are two problems that require major improvements before the paper can be considered for publication.

Comments:

Comment 1: 1. methodological issues
The authors put the emphasis on the effect of absorption profiles on the accuracy, but the data set does not really support this kind of research, because there is no information about food composition--note that not only the glycemic index is missing, the fiber and fat content also modifies the shape of the absorption curve. Experimenting with various absorption curve shapes for all patients and for all meals is a slight methodological error (no wonder it did not result in improvement) because the shape or the curve is determined by the meal much more than by the patient. Similarly, for the insulin profiles, the manufacturer data sheet should be used. In my opinion, this dataset would better support a study on the effect of physical activity on BGL prediction accuracy due to the good quality acc. data, i.e. training a model with and without this information. 
Another problem is that with 7 setups the focus of the paper is blurred, the reader is expected to be bored. The authors should select 2 or maximum 3 setups by dropping the trivial configs (e.g. those without food info). Selected models should be better analysed (e.g. check the effect of training data selection by cross validation). 
I also miss the discussion of the practical applicability of the method. The authors should mention that the method requires a CGM to be worn at all times, therefore the results are applicable only closed loop (CGM-insulin pump) setups.

Answer:

Thank you for the suggestions which have allowed us to improve the mansucript.

The absorption curves in the paper have used average values and are a first approximation that shows that there is a room for improvement with the previous studies based on raw signals. Using the approximate absorption curves have shown promising results for a single machine learning model when comparing with scenario 1 (which uses the raw data as in the majority of previous studies). Our intuition coincides with the reviewer in the sense that using a more complete meal dataset would allow us to use per meal absorption models which are expected to even further improve results. This does not mean that the contribution of the paper based on average signals is not sufficient to open this line of further research. We recognize that a limitation of the study is that we did not have all the food data that could have allowed us to implement per meal models and as future work we will apply the models to datasets with more complete data to be able to make a finer adjustment of the shape of the absorption curves.

In order to reduce the complexity of the explanations, some redundant data has been removed from the paper.

We have 7 different scenarios since it is important to add basic scenarios to be able to compare improvement when entering absorption curves. We think this provides relevant reference information to the paper since it allows us to comparte improvement with previous similar research studies.

We use training and cross-validation of samples per patient. As shown in the previous literature  the blood glucose prediction models better learn the particularities of each patient and do not generalize well between patients. We have used 10 different splits for training and validation for the data of each patient and average results have been computed for the results in the manuscript.

We have made clear that the model requires data from a  continuous blood glucose monitor (CGM) as recommended:

This paper uses a Recurrent Neural Network (RNN) based on LSTM cells in order to estimate future levels of blood glucose based on past readings coming from a continuous blood glucose monitor (CGM), insulin injections and carbohydrate intake and works on different absorption models to process the data available from real patients to obtain the best prediction of their next glucose levels and thus prevent possible rises or falls in the blood sugar level

Following the reviewer’s recommendation, we have extended the results to take into account the variation in results if no activity data is used. We have added an entire subsection to capture these details which, as the reviewer very well points at, are available in the dataset. The following content has been added to the manuscript:

[inserted as figure in the attached document]

Comment 2: 2. style and format of the text
The paper is too long and overly redundant. I estimate that half of the current volume would be enough for this content. Some points are repeated 3 or more times. Examples are sect. 2.4 and the lines 404-416, 331-336, 241-246. This will bore the readers.
Also, the style is superfluous and over-complicated over the whole text, e.g. lines 376-377 can be simply phrased as better accuracy. The style should be more formal cf. line 205 '...needed to do the five minutes'

Answer:

We have deleted some redundant information as suggested. The paper is more compact now.

Comment 3: Minor remarks

Remark: The nights should be separated from the data because of the lack of movement and glucose infusion (the modeling challenge is quite different).

Answer:

The following sentence was added “Windowing the input signals”.

“All the night measurements were eliminated due to the lack of acceleration data during that time.”

Remark: The main conclusions about the differences in accuracies should be statistically proven. The term 'significant' should not be used without such checks. Similary, the term 'optimized' would involve that the authors can prove there is no better solution (use 'improved' instead, also in the title).

Answer:

We have calculated the ANOVA test for assessing if the differences are statistically significant as suggested and introduced the following text in section 5.1:

The ANOVA test for the RMSE values for the 7 different experiments shows statistically significant differences (p-values under 0.05). In particular, the p-value is \[ 1.85 10^{-13} \], F-value is 12,98 and critical F-value is 2.12

We have used “improved” instead of “optimized” as proposed.

Remark: The lines 50-80 should separate the absorption model from the BGL regulation model.

Answer:

Thank you for the remark since it is important to position the absorption processes inside the entire BGL regulation models. We have added the following information in section 2:

Several absorption models for insulin and carbohydrates have been proposed inside blood glucose regulation models in order to assess the influence over time of human actions (meal intake or insulin injections) into plasma concentration levels and therefore provide an estimation of the influence in blood glucose levels.

Remark: In lines 90-120, focus on studies that are directly comparable to this study.

Answer:

Firstly,  an overview is given in order to show the use of machine learning in predicting blood glucose levels. However, as suggested, the final comparative analysis has been done with similar studies.

Remark: Table 1 should include only the 6 patients actually used, and be extended with information on gender, age, HgA1c, number of days and CGM records and lifestyle (inpatient or outpatient).

Answer:

The data is in the original publication in:

\bibitem[Author4(year)]{ref-dubosson}

  1. Dubosson et al., “The open D1NAMO dataset: A multi-modal dataset for research

on non-invasive type 1 diabetes management”, [On-line]. Available at:

https://www.diabetesatlas.org/upload/resources/material/20200302\_

133352\_2406-IDF-ATLAS-SPAN-BOOK.pdf.

Table 5 shows the patients of the Type-1 Diabetes subset (sorted by age).

[inserted image in the attached document]

  • A new (Table 2) is included with the Year, gender, height and Weight.
  • A paragraph is included: “There were 9 patients with normal outpatient lifestyles. Patients 003, 005 and 009 were not included in the study due to incompleteness in the meals and CGM data. Because of that, only 6 patients were included. Extended information about the relevant patients is shown in Table 2”.
  • Also, in Table 4 one column called “Total hours” is added. It shows the hours of the CGM record using the timestamp for the first and last measurement.

Remark: The accel. data is very valuable. However, it should be proven by argument or references that the stdev feature correctly depicts the volume/intensity of the activity.

Answer:

We have added a reference to a paper that uses activity data:

\bibitem[Author(year)]{ref-Georga} Georga, E. I., Protopappas, V. C., Polyzos, D., & Fotiadis, D. I.,  "Evaluation of short-term predictors of glucose concentration in type 1 diabetes combining feature ranking with regression models". Medical & biological engineering & computing, 2015, 53(12), 1305-1318.

We have added the following text to better describe the impact of activity data:

[inserted image in the attached document]

Remark: Include only information relevant to the study. In table 2, only Lispro and NPH should appear.

Answer:

It is right, so Determir and Glargina were deleted. Regular is shown because it is considered in the context.

Remark: The heurisic profiles used for glucose infusion do not resemble the curves of absorption models (the peak is not so pointed and sharp).

Answer:

The idea was to test several models and compare them.

Peak value and the duration have been adjusted and tested with different curves to be able to evaluate the effect of varying them on the data obtained.

The manufacturer's curves that are more similar to reality have also been used.

Remark: Table 3 should include the number of validation windows as well.

Answer: The table is augmented with new columns: “total windows”, “train windows”, “validation windows”, “test windows”. Also, the number of days with CGM samples per patient.

Remark: Table captions should provide a full explanation, e.g. in table 5, bold and underline should be explained.

Answer:

It is right, Tables 6 and 7 include a footnote.

Reviewer 2 Report

The present ms employs deep learning methods for predicting the glucose level. It is comprehensive and well-written. I have a few minor points:
1.    Title: The word ”Using“ is used twice. Maybe avoid this.
2.    Throughout the ms: What are ”real patients“ and ”normal humans“? Maybe these terms are not properly defined.
3.    Consider starting numbering the introduction with 1.
4.    L. 171 and others: The patient has the label ”001“, but in Table 1, it is only patient ”1“. Please use it consistently.
5.    L. 181ff.: Define the meaning ”acceleration data“.
6.    L. 207: Consider line break in ”30 000“
7.    L. 256, l. 257: V_{max} and K_m
8.    Equation 3: use ”\cdot“ instead of ”*“
9.    L. 338: what is ”4 o 5“?
10.    Section 5 is not an ordinary ”Discussion“. Is it maybe a summary subsection in the Results section? Also, Section 6 reads like a typical ”Discussion“ section instead of a ”Conclusion“ section.

Author Response

All the comments from the reviewers have been fully considered and implemented in order to improve the current version of the manuscript. We want to thank the reviewers for their feedback and comments that have allowed us to improve the quality of the paper.

Reviewer 2.

The present ms employs deep learning methods for predicting the glucose level. It is comprehensive and well-written. I have a few minor points:

Comments:
Comment 1.    Title: The word ”Using“ is used twice. Maybe avoid this.

Answer:

We have implemented the requested change in order to make the title more readable.

“Using Absorption Models for Insulin and Carbohydrates to Optimize Glucose Level Predictions using Deep Learning” has been changed to “Using Absorption Models for Insulin and Carbohydrates and Deep Learning to Improve Glucose Level Predictions”

Also, changed in “Citation:”.

Comment 2.    Throughout the ms: What are ”real patients“ and ”normal humans“? Maybe these terms are not properly defined.

Answer:

We have provided more information about what we mean by real patients the first time we use the term (people suffering Type 1 Diabetes Mellitus (T1DM)) and by normal humans (not suffering Type 1 Diabetes Mellitus (T1DM)).

Comment 3.    Consider starting numbering the introduction with 1.

Answer:

  1. Introduction → 1. Introduction

Comment 4.     L. 171 and others: The patient has the label ”001“, but in Table 1, it is only patient ”1“. Please use it consistently.

Answer:

We have changed Table 1 to Patient number: 001, 002… to be consistent with the rest of the manuscript

We have also changed “No dates available” → “no dates available”.

Comment 5.     L. 181ff.: Define the meaning ”acceleration data“.

Answer:

We have provided details about the origin of the acceleration data adding:

“Acceleration data, recorded from the accelerometer sensor worn by each patient,” or

“Acceleration data measures acceleration recorded from the accelerometer sensor worn by each patient. It is used to estimate if the patient performed physical activity in a time interval…”

Comment 6.     L. 207: Consider line break in ”30 000“

Answer:

We have deleted the line break.

Comment 7.    L. 256, l. 257: V_{max} and K_m

Answer:

We have implemented the subindexes as noted by the reviewer.

Comment 8.    Equation 3: use ”\cdot“ instead of ”*“.

Answer:

We have used “\cdot” instead of “*” in Equation 3.

Comment 9.     L. 338: what is ”4 o 5“?

Answer:

Changed “o” by “or”.

Comment 10.    Section 5 is not an ordinary ”Discussion“. Is it maybe a summary subsection in the Results section? Also, Section 6 reads like a typical ”Discussion“ section instead of a ”Conclusion“ section.

Answer:

The previous discussion section has been renamed as “Comparison with similar previous studies” following the recommendations of the reviewer.

The conclusions section has been renamed to “Discussion”.

Reviewer 3 Report

The paper is technically sound, well organized, and dedicated to an urgent topic.

The authors should consider using notorious models as the Bergman model (https://pubmed.ncbi.nlm.nih.gov/33658981/) and the Hovorka model (https://pubmed.ncbi.nlm.nih.gov/15382830/) to describe insulin and carbohydrates action or problems associated with the implementation of these models. The possibility of using detailed time-action profiles of insulins (such as those presented in L. Heinemann’s Time-action profiles of insulin preparations, ISBN: 3-87409-364-6 and in other publications of L. Heinemann) should be considered in the introduction/discussion.

The material and methods section would benefit if a more detailed CGM, insulin, and carbohydrates description of data would be presented: mean and SD for CGM values; insulin doses and meals. This data is essential when evaluating per patients variability in model accuracy.

Parts 3.1-3.7 and 4.1.1 – 4.1.7 of the article contain repetitive information and can be shortened without losing the lead.

The results section does not answer the question of accelerometer data plays any role in BG prediction, which is an exciting question.

According to the obtained results, the errors in data play a crucial role in prediction accuracy. The quality of carbohydrates data explained the significant difference in prediction accuracy between the patients. Hence, meal data flaw detection should be addressed at least briefly in the introduction/discussion. There were several proposals in recent publications to detect false meal data by patients (e.g., https://www.ncbi.nlm.nih.gov/pmc/articles/PMC5867514/, https://ieeexplore.ieee.org/document/9281297). The impact of a proper flawed meal data detection algorithm on the preprocessing stage might be crucial.

The results section could benefit if R values (correlation between real and predicted values) were estimated together with RMSE, as those are frequently used for blood glucose prediction models.

The discussion section could benefit from considering that the more complex models would not require the exponential models?

Minor edits:

Line 330. Misspelled 4 or 5.

Line 696. Misspelled RMSE.

Line 747. Misspelled CGM.

Author Response

All the comments from the reviewers have been fully considered and implemented in order to improve the current version of the manuscript. We want to thank the reviewers for their feedback and comments that have allowed us to improve the quality of the paper.

Reviewer 3.

The paper is technically sound, well organized, and dedicated to an urgent topic.

Comments:
Comment 1.    The authors should consider using notorious models as the Bergman model (https://pubmed.ncbi.nlm.nih.gov/33658981/) and the Hovorka model (https://pubmed.ncbi.nlm.nih.gov/15382830/) to describe insulin and carbohydrates action or problems associated with the implementation of these models. The possibility of using detailed time-action profiles of insulins (such as those presented in L. Heinemann’s Time-action profiles of insulin preparations, ISBN: 3-87409-364-6 and in other publications of L. Heinemann) should be considered in the introduction/discussion.

Answer:

We want to thank the reviewer for the recommendation. In fact, we fully agree and we have dedicated the first paragraph in section 2 to capture major mathematical models to explain the glucose-insulin regulation in the human body. The reference was already explained in the paper and the new one (https://pubmed.ncbi.nlm.nih.gov/33658981/)  has been added together with the following text:

An explanatory review about early mathematical models designed to understand the pathogenesis of diabetes is captured in [10]

 The reference “L. Heinemann’s Time-action profiles of insulin preparations” has been added and the curves proposed in that reference have been linked to those used in the paper from a different reference. We have added the text:

From the data in the Table2, action curves of the different types of insulin shown in the Figure 5 have been generated \cite{ref-wang}. Similar models for time-action profiles of insulin preparations are captured in  \cite{ref-Heinemann}

Comment 2.    The material and methods section would benefit if a more detailed CGM, insulin, and carbohydrates description of data would be presented: mean and SD for CGM values; insulin doses and meals. This data is essential when evaluating per patients variability in model accuracy.

Answer:

The data is in the original publication in:

\bibitem[Author4(year)]{ref-dubosson}

  1. Dubosson et al., “The open D1NAMO dataset: A multi-modal dataset for research

on non-invasive type 1 diabetes management”, [On-line]. Available at:

https://www.diabetesatlas.org/upload/resources/material/20200302\_

133352\_2406-IDF-ATLAS-SPAN-BOOK.pdf.

Table 6 shows the per-patient statistics of glucose measurements.

[image added, please see the attached document]

The following text has been added to the manuscript:

The individual glucose statistics for the 9 patients are captured in Figure 6 in \cite{ref-dubosson}.

Comment 3.    Parts 3.1-3.7 and 4.1.1 – 4.1.7 of the article contain repetitive information and can be shortened without losing the lead.

Answer:

Redundant information has been deleted from subsections 4.1.1 to 4.1.7 as requested.

Comment 4.     The results section does not answer the question of accelerometer data plays any role in BG prediction, which is an exciting question.

Answer:

A new subsection has been added to answer the question of acceleration data plays any role in BG prediction as suggested. The following information has been added:

[image added, please see the attached document]

Comment 5.     According to the obtained results, the errors in data play a crucial role in prediction accuracy. The quality of carbohydrates data explained the significant difference in prediction accuracy between the patients. Hence, meal data flaw detection should be addressed at least briefly in the introduction/discussion. There were several proposals in recent publications to detect false meal data by patients (e.g., https://www.ncbi.nlm.nih.gov/pmc/articles/PMC5867514/, https://ieeexplore.ieee.org/document/9281297). The impact of a proper flawed meal data detection algorithm on the preprocessing stage might be crucial.

Answer:

The reviewer is completely right. We ourselves are developing and tuning machine learning based methods for food intake estimations but it will be presented as a future work. We have captured the references proposed and the importance of the recommendation adding the following text to section 3:

The information in the D1NAMO dataset \cite{ref-dubosson} is likely not to be complete. Meals and insulin injections were manually recorded by the participants and it is likely that some of the data values for some meals and insulin injections are missing. \hl{Several studies have highlighted the importance of having accurate data for meals. The study in} \cite{ref-Samadi} \hl{analysed CGM postprandial data and using fuzzy logic the authors were able to detect the majority of meals and many of the snacks. The authors in} \cite{ref-Pustozerov} \hl{also used a set of rules to detect incorrect meal records. As a future work, we plan to use machine learning techniques to better detect missing or incorrect information in the annotated meals.}

Comment 6.     The results section could benefit if R values (correlation between real and predicted values) were estimated together with RMSE, as those are frequently used for blood glucose prediction models.

Answer:

The author is right and there are some previous research studies that use the Pearson's correlation coefficient in order to assess the accuracy of predictions. However, the RMSE value has been used in all previous studies and it has been convenient for us in order to do a comparison with them. We have captured a reference in which the R value is used and we have added the following text in order to provide a justification for the chosen measures:

“RMSE values have been used in order to be able to compare results with previous studies which have published these values for their models. Another parameter that could be added in order to assess the accuracy of predictions is the Pearson's correlation coefficient such as in  \cite{ref-Uenaka}.”

Comment 7.    The discussion section could benefit from considering that the more complex models would not require the exponential models?

Answer:

 We have captured the idea and added the following information to the discussion of results following the reviewer’s recommendation:

\hl{Using absorption curves, as proposed in this paper, the intuition is that the input data to the machine learning model reduces the complexity of the patterns to be learnt as compared to models such as} \cite{ref-munoz-organero} \hl{which propose deeper architectures that try to learn absorption curves based on raw data. Deeper models require more data to be trained which in some cases could require T1DM patients to have to train the system for a longer period of time before being able to use it.}

Comment 8.    Minor edits:

Line 330. Misspelled 4 or 5.

Line 696. Misspelled RMSE.

Line 747. Misspelled CGM.

Answer:

Changed “o” by “or.

Changed “RSME” by “RMSE”.

Changed “GCM” by “CGM”.

Round 2

Reviewer 1 Report

The paper has been significantly improved. An issue that should still be resolved is that you use the units mg/dl and mmol/l units incoherently: in some cases only mg/dl, in others only mmol/l and in Table 8 both. You should choose one and state clearly at the beginning of the paper that "for clarity, throughout this paper the unit mmol/l will be used for BGL etc.". Otherwise, you must explicitly use the unit after each numerical value.

There are still formatting errors, e.g. Sect. 5.2 begins with "In Figure ?? the boxplots..." Please check the whole text again.

Author Response

All the comments from the reviewers have been fully considered and implemented in order to improve the current version of the manuscript. We want to thank the reviewers for their feedback and comments that have allowed us to improve the quality of the paper.

Reviewer 1.

The paper has been significantly improved.

Comments:

Comment 1: An issue that should still be resolved is that you use the units mg/dl and mmol/l units incoherently: in some cases only mg/dl, in others only mmol/l and in Table 8 both. You should choose one and state clearly at the beginning of the paper that "for clarity, throughout this paper the unit mmol/l will be used for BGL etc.". Otherwise, you must explicitly use the unit after each numerical value.

Answer:

-In “Data Pre-Processing”, the following sentence is added: “For clarity, throughout this paper the unit mmol/L  is used for Blood Glucose Prediction.”

-In Table 6, the unit is added. “Obtained results of each experiment number” ->  “Obtained results of each experiment number in mmol/L”

-In “5.1 Results per experiment”, the unit is added. “Each experiment aggregates the RMSE values for the 10 executions for the 6 participants.”-> “Each experiment aggregates the RMSE values in mmol/L for the 10 executions for the 6 participants.”

-In Table 7, the unit is added. “Obtained results for experiment 1 with and without data” ->  “Obtained results for experiment 1 with and without data in mmol/L”

-In “Comparison with similar previous studies”, the “Output error values” paragraph is extended with  “In the present work the values are in mmol/L. However, the values are converted into mg/dL because this unit is used more often in previous studies.”

-In “Comparison with similar previous studies”, the following paragraph is added.  “In this work, the RMSE values have been computed with glucose levels in mmol/L. However, it is important to convert the units to mg/dL. In case of the first experiment, the result is 0.863 (mmol/L) = 15.5 (mg/dL). The result is improved in the fifth experiment with a mean value of 0.510 (mmol/L) = 9.2 (mg/dL).”

 -In Table 8, the values in mmol/L are deleted in Experiment 1 and Experiment 5.

Comment 2: There are still formatting errors, e.g. Sect. 5.2 begins with "In Figure ?? the boxplots..."

Answer:

As suggested, the reference in Figure 21 is fixed.

Minor remarks: There are still formatting errors. Please check the whole text again.

Remark: In 5.1 Results per experiment, “the p-value is 1.8510^-13”, mathematical delimiters were changed. 

Answer:

Before, it was in unnumbered displayed mode with delimiters \[\].

Now, it is in inline mode using the delimiters \(\).

Remark: In 5.1 Results per experiment, the decimal separator is changed into “.” .

Answer: “F-value is 12,98” -> “F-value is 12.98”.

Remark: In 5.1 Results per experiment, a final dot is added .

Answer: “and critical F-value is 2.12” -> “and critical F-value is 2.12.”

Remark:  In 5.1 Results per experiment, the following sentence is changed.

Answer:

“In Figure 11 a boxplot is used in order to show the achieved results.” ->

“In order to show the achieved results, a boxplot in Figure 11 is included.”

Reviewer 3 Report

The manuscript has been sufficiently improved. All the highlighted issues were addressed by the authors and necessary information was added to the manuscript.

Author Response

Reviewer 3.

The manuscript has been sufficiently improved. All the highlighted issues were addressed by the authors and necessary information was added to the manuscript.

Answer:

We want to thank the reviewer for his/her previous comments that have allowed us to improve the manuscript.
